# The origin and degassing history of the Earth's atmosphere revealed by Archean xenon

Guillaume Avice[1,†], Bernard Marty[1] & Ray Burgess[2]

Xenon (Xe) is an exceptional tracer for investigating the origin and fate of volatile elements on Earth. The initial isotopic composition of atmospheric Xe remains unknown, as do the mechanisms involved in its depletion and isotopic fractionation compared with other reservoirs in the solar system. Here we present high precision analyses of noble gases trapped in fluid inclusions of Archean quartz (Barberton, South Africa) that reveal the isotopic composition of the paleo-atmosphere at $\approx 3.3$ Ga. The Archean atmospheric Xe is mass-dependently fractionated by $12.9 \pm 2.4 \text{‰ u}^{-1}$ ( $\pm 2\sigma$, s.d.) relative to the modern atmosphere. The lower than today [129]Xe excess requires a degassing rate of radiogenic Xe from the mantle higher than at present. The primordial Xe component delivered to the Earth's atmosphere is distinct from Solar or Chondritic Xe but similar to a theoretical component called U-Xe. Comets may have brought this component to the Earth's atmosphere during the last stages of terrestrial accretion.

[1] CRPG-CNRS, Université de Lorraine, UMR 7358, 15 rue Notre-Dame des Pauvres, BP 20, 54501 Vandoeuvre-lès-Nancy Cedex, France. [2] School of Earth and Environmental Sciences, University of Manchester, Oxford Road, Manchester M13 9PL, UK. † Present address: Division of Geological and Planetary Sciences, California Institute of Technology, Pasadena, California 91125, USA. Correspondence and requests for materials should be addressed to G.A. (email: gavice@caltech.edu).

The building blocks of the Earth accreted in regions where temperatures were too high to permit significant retention of volatile elements (for example, H,C,N, noble gases). The Earth probably acquired its volatile components late from more distant sources[1] such as the Main Belt asteroids and/or comets. Simulations suggest that during the final stages of solar system formation, the orbits of asteroids were disturbed by the migration of the giant planets and, for some of them, their trajectories crossed the Earth's orbit[2]. Even if the isotopic composition of cometary water argues against comets being the source of terrestrial water[3], the high $Ar/H_2O$ ratio measured in comet 67P/C-G by the Rosetta spacecraft[4] suggests that cometary objects may have contributed noble gases to the terrestrial atmosphere[5].

At least two distinct cosmochemical sources contributed noble gases to the Earth's mantle: a Solar end-member detected in Ne isotopes[6] and a Chondritic component (that is, asteroidal) apparent in the isotopic compositions of Kr (ref. 7) and Xe (ref. 8). The presence of radiogenic and fissiogenic noble gases in the atmosphere ($^{40}Ar$, $^{129,131-136}Xe$), produced by the radioactive decays of parent nuclides ($^{40}K$, $^{129}I$, $^{244}Pu$, $^{238}U$) in the mantle and crust, attests for exchanges between the surface and the silicate Earth. However, the ultimate origin of the Earth's atmosphere remains unknown, especially for xenon. The Xe abundance in the Earth's atmosphere is depleted, the atmospheric Xe/Kr ratio being lower by a factor of $\sim 20$ relative to the chondritic composition[9,10]. Furthermore, atmospheric Xe is enriched in heavy isotopes by $30-40\text{‰}\,u^{-1}$ relative to Chondritic (Q-Xe) or Solar (SW-Xe) (ref. 11). These two features form the so-called 'xenon paradox'[12]. When corrected for mass-dependent isotope fractionation, atmospheric Xe is depleted in its heavy isotopes ($^{134}Xe$ and $^{136}Xe$) relative to Solar or Chondritic Xe, and cannot be related to any known cosmochemical component[13,14]. These observations led to the definition of a theoretical primordial component labelled 'U-Xe' (ref. 13), which has solar-like composition for the light isotopes $^{124-130}Xe$, and is depleted in heavy Xe isotopes. However, this composition was derived from statistical correlations and its presence has never been observed in any terrestrial or extraterrestrial material. Recently, Meshik et al.[15] proposed an alternative explanation to U-Xe for the origin of atmospheric Xe. Their model is not based on a different primordial component but involves Chemically Fractionated precursors of Fissiogenic Xe producing the so-called CFF-Xe (see ref. 16 and refs therein). Degassing of this component, enriched in $^{134}Xe$ and $^{136}Xe$, accompanied by atmospheric escape seems to match the fission spectrum of atmospheric Xe. However, when SW-Xe is taken as a precursor of atmospheric Xe and is mass-fractionated to reproduce, for example, the $^{128}Xe/^{130}Xe$ in the modern atmosphere, the result produces anomalous $^{136}Xe$ excesses (about $26 \pm 6\text{‰}$ ($1\sigma$) compared with the modern atmosphere) before any fission contribution. Thus, CFF-Xe cannot be the sole explanation for the origin of the isotopic composition of atmospheric Xe since this modified fission component can only increase the $^{136}Xe$ budget of the atmosphere.

Recent studies of Archean barite and quartz samples from North Pole, Pilbara (NW Australia) demonstrated that, 3.5 to 3.0 Ga ago, atmospheric Xe had an isotopic composition less isotopically fractionated than the modern atmospheric Xe relative to any of the potential primordial components[17-20]. These data suggest a progressive long-term evolution of the isotopic composition of atmospheric Xe by mass-dependent isotope fractionation, that may be due to ionization of atmospheric xenon[21] by ancient, ultraviolet-rich, solar radiation and progressive escape of Xe ions to space[22,23]. However, these studies did not elucidate the original composition of atmospheric xenon, which was then tentatively attributed to Solar/

Chondritic[10,19]. Xenon escape processes could have also led to mass-independent isotope fractionation, in addition to the mass-dependent one, that could account for the unique isotope composition of modern atmospheric Xe. To investigate the ultimate origin of atmospheric xenon, we selected and analysed with unprecedented high precision Archean quartz samples from the Barberton Greenstone Belt (BGB), South Africa.

Here we demonstrate that fluid inclusions in the Barberton quartz samples record the Xe isotope composition of the Archean atmosphere. This 3.3 Ga-old atmospheric Xe is mass-dependently fractionated by $\approx 13\text{‰}\,u^{-1}$. Depletion in radiogenic $^{129}Xe$ relative to the modern atmosphere allows us to compute a degassing rate from the Earth's mantle to the atmosphere over the last 3.3 Ga. Furthermore, Archean Xe originates from a primordial component different from all other known reservoirs of Xe in the solar system and similar to the theoretical U-Xe. Comets may have been the source of this noble gas component, added to the Earth's atmosphere during late accretionary events.

## Results

**Samples characteristics**. Samples analysed in this study are from a core (BARB 3) drilled in the BGB, South Africa. The drilling project is part of an ICDP Project ('Peering into the cradle of life', PI: N. Arndt). The BARB 3 core was drilled in rocks of the Kromberg formation (3.33–3.47 Ga) and mainly comprises a succession of white and black cherts and ultramafic rocks[24]. All samples of this study consist of macro-crystalline quartz (Supplementary Fig. 1) with different modes of emplacement in rocks from the BGB and probably linked to early hydrothermal activity[25]. Some of the samples are from well-defined cm-sized bedded veins (Supplementary Fig. 1) with sharp, straight (Supplementary Fig. 1a) or irregular (Supplementary Fig. 1b) contacts with adjacent white chert; other samples are from meter-sized coarse quartz veins. Analytical techniques used during the $^{40}Ar$-$^{39}Ar$ experiments and Xe-Kr analyses are described in Methods.

**$^{40}Ar$-$^{39}Ar$ ages**. Results of the $^{40}Ar$-$^{39}Ar$ crushing experiments (Methods, Supplementary Table 1) show anomalously high ages, often exceeding the age of the solar system ($>4.56$ Ga), due to the presence of $^{40}Ar$ excess ($^{40}Ar_E$) (ref. 26). This excess argon component is correlated with the chlorine (Cl) content (Figs 1 and 2) and probably reflects enrichment of the fluids in crustal-derived radiogenic argon and chlorine during fluid-rock interaction processes before entrapment. Only step-heating steps yielding elevated K/Cl ratios give realistic ages between 3 and 3.5 Ga that are broadly compatible with the age of the formations in which the quartz veins are emplaced (Fig. 2). We correct $^{40}Ar$-$^{39}Ar$ data for excess argon using a plane ($R^2 = 0.975$) to the data in three-dimensional (3D) space (Cl-K-$^{40}Ar$) space by applying a Monte Carlo method to propagate errors on measurements (Methods, Fig. 3 and Supplementary Fig. 2). This approach leads to a $^{40}Ar$-$^{39}Ar$ age of $3.3 \pm 0.1$ Ga ($\pm 2\sigma$, s.d.) for Barberton sample BMGA3-9 (Fig. 3). This age is similar within errors to the formation age[24] but may also be consistent with late hydrothermal fluid circulation events[25] linked to the intrusion of adjacent 3.22 Ga-old plutons[27]. The initial $^{40}Ar/^{36}Ar$ is $458 \pm 4$ ($\pm 2\sigma$, s.d.) for sample BMGA3-9, which is higher than the modern atmospheric ratio of 298.6 (ref. 28), this may be explained by the presence of some $^{40}Ar$ excess uncorrelated with the chlorine content.

A different approach (Methods) was used to constrain the age of sample BMGA3-13, which contains relatively lower excess $^{40}Ar$ abundances as indicated by a less-well defined correlation between $^{40}Ar_E$ and Cl ($R^2 = 0.95$) and lower $^{40}Ar/^{36}Ar$ values

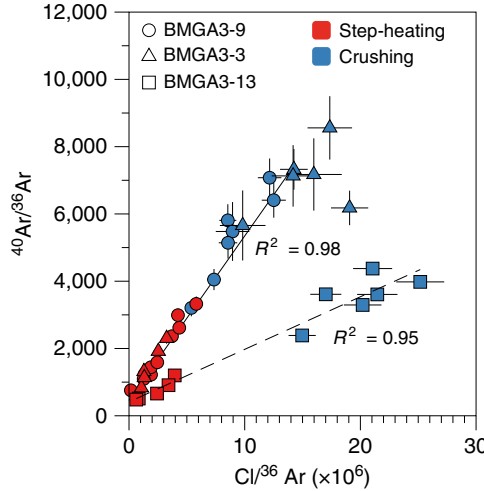

**Figure 1 | Ar-Cl correlation for results of crushing and step-heating experiments on samples BMGA3-9, BMGA3-13 and BMGA3-3.** The correlation demonstrates the presence of $^{40}Ar$ ($^{40}Ar_E$) excess related to the chlorine content. $^{40}Ar$ excess prevents the direct determination of an age from any simple $^{40}Ar$-K correlation. Samples BMGA3-9 and BMGA3-3 show a similar correlation of $^{40}Ar$ with chlorine content (solid line). Sample BMGA3-3 contains lower argon excess $^{40}Ar$ (dashed line). Regression lines and their determination coefficients ($R^2$) are indicative of the $^{40}Ar$-Cl correlations and are not error-weighted. Error bars at 1σ.

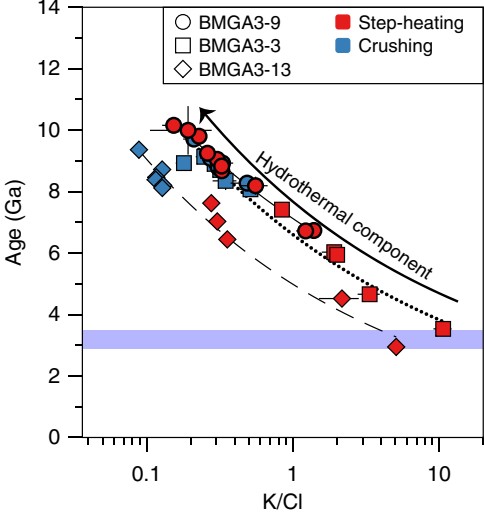

**Figure 2 | Apparent ages versus the K/Cl ratio.** Most ages are anomalously high (> 4.56 Ga) due to the presence of excess argon linked to a hydrothermal component rich in Cl (low K/Cl). For elevated K/Cl ratio (>3) apparent ages decrease towards more realistic values between 3 and 3.5 Ga (purple range). The solid, dotted and dashed curves schematically represent the evolution of ages with the K/Cl ratio for samples BMGA3-3, BMGA3-9 and BMGA3-13, respectively. Errors bars are at 1σ.

(Fig. 1 and Supplementary Table 1). This method uses the $^{40}Ar_E$/Cl derived from crushing data to correct step-heating data for the $^{40}Ar_E$ component (Methods). This leads to a similar, but less precise, age of 3.5 ± 1.0 Ga (± 2σ, s.d., mean square weighted deviation (MSWD) = 1.06). The initial atmospheric $^{40}Ar/^{36}Ar$ ratios of sample BMGA3-13, computed for ages varying between 3.2 and 3.4 Ga (see above) have values ranging from 178 to 202 with a mean of 190 ± 12 (± 2σ, s.d.) (Supplementary Fig. 3). This value is a minimum value for the Archean atmospheric $^{40}Ar/^{36}Ar$

ratio but is in broad agreement with previous estimates and prediction (143 ± 21, 3.5 Ga ago, (ref. 20)) and models invoking a peak in crustal extraction between 3.8 and 2.5 Ga (refs 20,29).

**Xenon isotopic composition.** The reproducibility of the crushing experiment results (Supplementary Table 2) on different samples and duplicates (Methods and Supplementary Fig. 4) enables a precise determination of the error-weighted average for the isotopic ratios of xenon trapped in Barberton quartz (Fig. 4 and Supplementary Table 2). The isotopic spectrum of xenon in Barberton quartz normalized to $^{130}Xe$ indicates excesses of the light isotopes ($^{124-129}Xe$) together with depletions of heavy isotopes ($^{131-136}Xe$) relative to the modern atmospheric composition (Fig. 4a). The absence of mantle-derived $^{129}Xe$ excesses from the decay of now extinct $^{129}I$ ($T_{1/2} = 16$ Ma) relative to the atmospheric composition (Fig. 4), together with an isotopic composition of krypton similar to the modern atmosphere (Fig. 5) argues against the presence of a mantle-derived component trapped within the fluid inclusions. Xenon in Barberton quartz thus has an Archean isotopic composition that differs from the modern atmosphere. The isotopic fractionation of xenon in Barberton quartz relative to the isotopic composition of the modern atmosphere was computed using the light stable, non-fissiogenic, non-radiogenic isotopes of Xe ($^{126,128,130}Xe$) plus $^{131}Xe$, for which production by the fission of $^{238}U$ is small[30]. Error-weighted correlations were obtained using the Isoplot 4.1 software[31], however, note that correlations involving $^{124}Xe$ were excluded for reasons explained in the Supplementary Discussion (Supplementary Information). The results show that Archean atmospheric xenon was isotopically fractionated by 12.9 ± 2.4‰ u$^{-1}$ (± 2σ s.e.m., MSWD = 1.4) relative to modern atmospheric Xe (Fig. 6).

**Discussion**
Our observations provide strong confirmation, with greater precision, of previous observations on samples from different regions and geological settings (Fig. 6)[17-20,32] that the peculiar Xe isotopic composition during the Archean eon was ubiquitous and not due to local fractionation effects. The results provide confirmation for the specific and long-term evolution of the isotopic composition of atmospheric Xe, since the fractionation determined here is significantly different from 21 ± 6‰ u$^{-1}$ (± 2σ, s.d.) Xe in North Pole, Pilbara barite with an age of 3.48 ± 0.18 Ga (± 2σ, s.d.)[18] (Fig. 6). A mass-dependent isotope fractionation process acted on atmospheric Xe isotopes from at least 3.3 Ga until the present day. However, neither modern atmospheric Xe, nor Archean atmospheric xenon trapped in Barberton quartz can be derived from Q-Xe (Chondritic) or SW-Xe (Solar) by mass-dependent isotope fractionation, because these primordial components carry $^{134,136}Xe$ excesses[13] relative to ancient atmospheric Xe (Figs 7 and 8; see also Supplementary Fig. 5). In the following discussion, we show how the Archaen atmosphere composition is consistent with U-Xe being a primordial component present in the atmosphere ≥ 3.3 Ga ago.

We start by making the reasonable assumption that the primordial component had a solar-like $^{132}Xe/^{130}Xe$ ratio of 6.061 ± 0.029 (± 2σ, s.d.)[33] (Fig. 7a, Supplementary Discussion); this is supported by the following: (1) Barberton $^{126-132}Xe/^{130}Xe$ (and not $^{134-136}Xe/^{130}Xe$) ratios can be related to solar Xe by a mass-dependent isotopic fractionation of ≈ 25‰ u$^{-1}$; (2) Solar gas represents the major gas reservoir of the solar system and any higher $^{132}Xe/^{130}Xe$ ratios as found in Chondritic meteorites are often accompanied by $^{134}Xe$ and $^{136}Xe$ excesses[34]. Note that this starting assumption is valid for $^{132}Xe$ only and does not imply that the whole isotopic spectrum of primordial Xe component

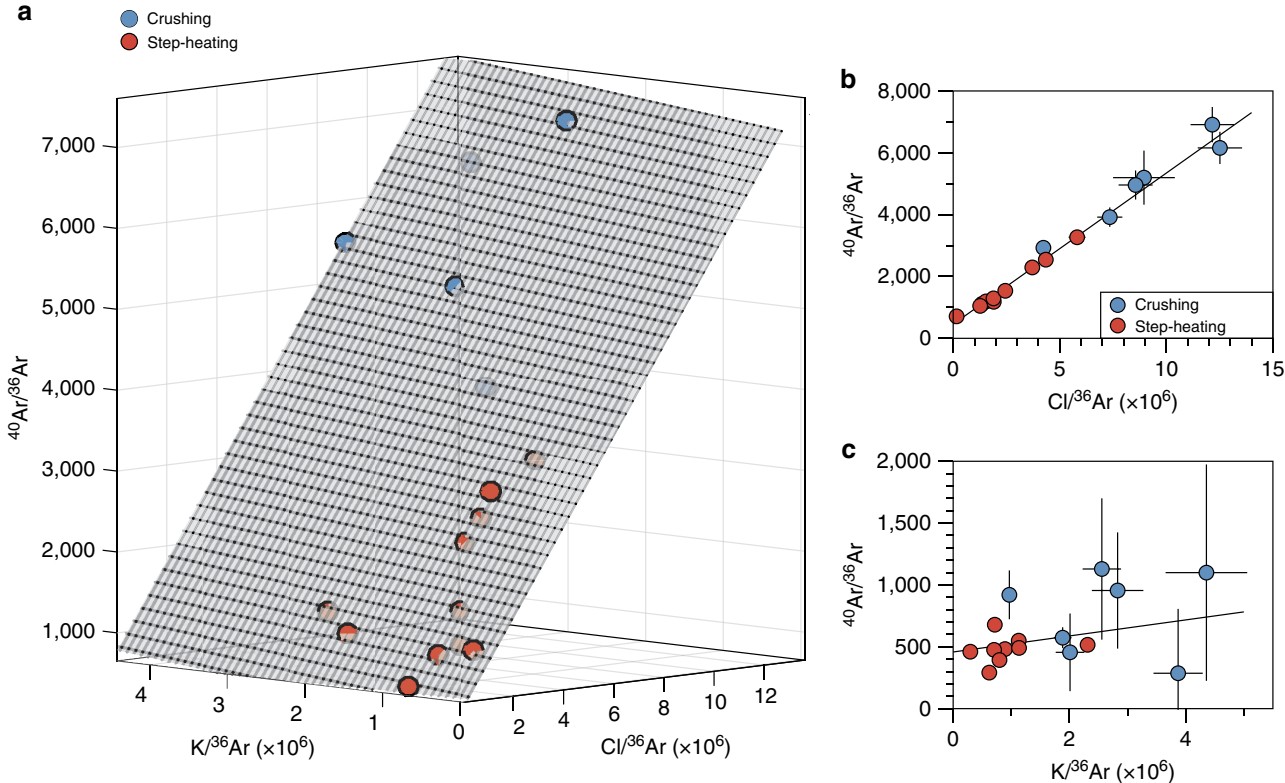

**Figure 3 | ⁴⁰Ar-K-Cl-³⁶Ar multi-component diagrams.** Results for crushing steps are shown with blue-filled circles and those for step-heating steps are shown with red-filled circles. (**a**) Three-dimensional representation of the ⁴⁰Ar-³⁹Ar data for sample BMGA3-9 in the ⁴⁰Ar-K-Cl space. The fitted plane (Methods) is shown as a grey mesh. (**b**) Edge-on view of the plane (black line) and data in the ⁴⁰Ar-Cl space. (**c**) Edge-on view of the plane (black line) and data in the ⁴⁰Ar-K space. Errors are at 1σ.

corresponds to Solar Xe. The heavy isotopes of xenon, for example, $^{134}$Xe and $^{136}$Xe, have been produced over time by spontaneous fission of $^{238}$U ($T_{1/2} = 4.47$ Ga) and $^{244}$Pu ($T_{1/2} = 80$ Ma). $^{244}$Pu was totally extinct $\approx 4.1$ Ga ago, and only the spontaneous fission of $^{238}$U has contributed Xe isotopes to Barberton fluid inclusions from 3.3 Ga ago to the present (Fig. 7b). This contribution could have taken place either *in situ* in fluid inclusions or in the surrounding crustal rocks leached by the fluids before their entrapment in Barberton quartz samples. Light isotopes excesses in the Barberton Xe isotopic spectrum (Fig. 4) correspond to an isotopic fractionation ($\approx 12.9$ ‰ u$^{-1}$) that can be propagated towards heavy isotopes to compute a theoretical primordial isotopic composition for the atmosphere, especially for $^{134}$Xe and $^{136}$Xe (Fig. 7c). The resulting range of possibilities for the initial isotopic composition (range at $\pm 2\sigma$) corresponds to $^{136}$Xe/$^{130}$Xe $= 1.685 \pm 0.075$ ($\pm 2\sigma$, s.d.) (Figs 7d and 8) and $^{134}$Xe/$^{130}$Xe $= 2.14 \pm 0.07$ ($\pm 2\sigma$, s.d.) (Methods, Supplementary Fig. 5). The mass-dependent isotopic fractionation of this composition, combined with the addition of fissiogenic $^{132-136}$Xe isotopes with known yields[12], accounts for the Xe isotopic composition measured in Barberton quartz (Fig. 8 and Supplementary Fig. 5). Hence, our results demonstrate that an initial isotopic composition different from Q-Xe or SW-Xe, and similar to U-Xe, must have existed in the ancient atmosphere, without making use of the isotopic composition of the modern atmosphere as it was done in previous studies[13,14]. The fidelity of our deconvolution of Archean Xe is revealed by the Xe fission spectra shown in Fig. 9 where $^{131-136}$Xe excesses, following correction for mass-dependent fractionation relative to U-Xe, perfectly match the Xe spectrum for the spontaneous fission of $^{238}$U (ref. 30). Other potential compositions (Solar or Chondritic) lead to fission spectra neither related to the

spontaneous fission of $^{238}$U nor to that of $^{244}$Pu. A small contribution of the spontaneous fission of $^{244}$Pu in the total Xe fission component cannot be rejected. However, this contribution must be small (max. 2% based on the $^{134}$Xe/$^{136}$Xe ratio) and the abundances of $^{131-136}$Xe derived from the spontaneous fission of $^{244}$Pu are too low to be estimated accurately due to the overwhelming dominance of fissiogenic Xe isotopes from spontaneous fission of $^{238}$U.

The modern atmosphere contains $(4.06 \pm 0.05, \pm 1\sigma) \times 10^{12}$ mol of $^{129}$Xe (ref. 12) comprising a radiogenic $^{129}$Xe excess ($^{129}$Xe(I) hereafter) of $6.8 \pm 0.3\%$ ($\pm 1\sigma$, s.d.) (ref. 35) produced by the radioactive decay of extinct $^{129}$I ($T_{1/2} = 16$ Ma). For the Archean atmosphere, the excess of $^{129}$Xe(I), is calculated to be $6.07 \pm 0.22\%$ ($\pm 1\sigma$, s.d.) after correction for isotopic fractionation (Methods). This value is lower than the modern atmospheric excess (Fig. 4b) and probably results from the amount of $^{129}$Xe(I) that has been degassed from the mantle[12] during the last 3.3 Gyr. Thus, by comparing the $^{129}$Xe/$^{130}$Xe of 3.3 Ga-old atmosphere to that of the modern atmosphere provides insight into mantle degassing, and, therefore, convection, through time. This approach is advantageous compared with atmospheric $^{40}$Ar degassing models because the latter are much more closely related to crustal production than mantle degassing flux (for example, ref. 20), due to $^{40}$K being dominantly stored in the continental crust. The atmospheric increase in $^{129}$Xe(I) excess between 3.3 Ga and the present day corresponds to an integrated $^{129}$Xe(I) degassing rate of $8 \pm 4$ mol a$^{-1}$ ($\pm 1\sigma$, s.d.) (Methods). The modern mantle degassing rate of $^{129}$Xe(I) can be estimated from the $^{3}$He anomaly at mid-ocean ridges (MOR) and the mantle $^{130}$Xe/$^{3}$He value in MOR basalts (MORB) and continental well gases. The present degassing flux of $^{3}$He from the upper-mantle at MOR

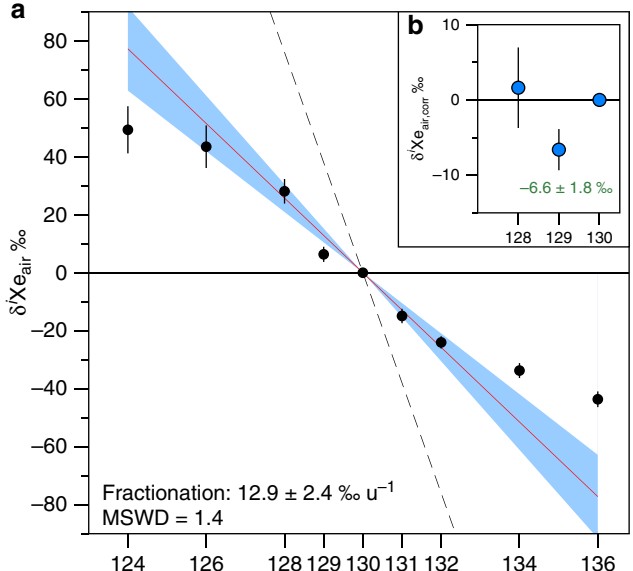

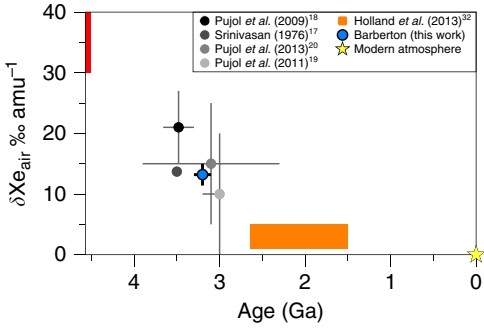

**Figure 4 | Isotopic composition of xenon in Barberton quartz samples.**
(**a**) Isotope spectrum of xenon in Barberton quartz samples (error-weighted average on 27 measurements) relative to the isotopic composition of the modern atmosphere and expressed using the delta notation $(\delta^i Xe_{air} = ((^i Xe/^{130}Xe)_{Barb}/(^i Xe/^{130}Xe)_{air} - 1) \times 1,000)$. The computed isotopic fractionation ($12.9 \pm 2.4\,‰\,u^{-1}$ ($\pm 2\sigma$)) is shown as a red line with its $2\sigma$ error envelope in blue. The dashed line corresponds to the isotopic fractionation of SW-Xe relative to the air ($38\,‰\,u^{-1}$, ref. 33). MSWD is mean square weighted deviation. Note the depletion in $^{129}Xe$ relative to the mass fractionation corresponding to lower radiogenic $^{129}Xe$ excess in the Archean atmosphere. Errors at $2\sigma$. (**b**) Isotope spectrum of $^{128}Xe$ and $^{129}Xe$ normalized to the isotopic composition of the modern atmosphere (see above) in Barberton quartz samples and corrected for the isotopic fractionation of $12.9 \pm 2.4\,‰\,u^{-1}$ ($\pm 1\sigma$, s.e.m.) ($\delta^i Xe_{air,corr}$). The $^{129}Xe$ depletion ($-6.6 \pm 1.8\,‰$ ($\pm 1\sigma$, s.e.m.)) corresponds to a degassing rate of $8 \pm 4$ ($\pm 1\sigma$, s.d.) $mol\,a^{-1}$ of radiogenic $^{129}Xe$ produced by the decay of now extinct $^{129}I$ ($T_{1/2} = 16\,Ma$). Errors at $1\sigma$.

**Figure 6 | Isotopic fractionation relative to the modern atmosphere of atmospheric Xe with time.** Isotopic composition is expressed in per mil per atomic mass unit (‰ u$^{-1}$) relative to the modern atmosphere (yellow star). Data for fractionated Xe compositions from the literature are indicated by the solid black and grey circles (refs 17–20) and by the orange range[32]. Starting isotopic fractionation (red range) varies between 30 and 40 ‰ u$^{-1}$ depending if it is Solar (SW-Xe)/U-Xe or Chondritic (Q-Xe) (ref. 11). Data obtained in this study for Barberton samples are indicated with a solid blue circle and correspond to the isotopic fractionation computed with the Isoplot software taking the error-weighted average on all 27 measurements. Errors at $2\sigma$ for literature data and data from this work. No error is given for the fractionation described in ref. 17.

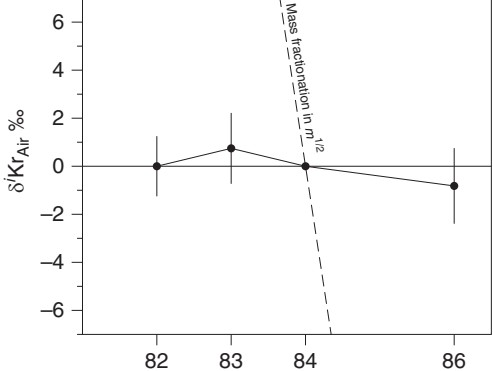

**Figure 5 | Isotopic ratios of krypton in fluid inclusions of Barberton quartz crystals.** Isotopic ratios are expressed with the delta notation relative to $^{84}Kr$ and to the isotopic composition of the modern atmosphere. The dashed line represents the isotopic fractionation measured for Xe ($13\,‰\,u^{-1}$) propagated towards Kr following a mass fractionation law proportional to $m^{1/2}$. Errors at $2\sigma$.

is estimated at $527 \pm 102\,mol\,a^{-1}$ (ref. 36). Estimates for the mantle $^{130}Xe/^{3}He$ ratio range from $0.85 \times 10^{-3}$ to $3.5 \times 10^{-3}$ (refs 37–39) and values between 1.29 and 1.7 for the $^{129}Xe(I)/^{130}Xe$ (ref. 40), this leads to a modern degassing rate of

$1.37 \pm 0.88$ ($\pm 1\sigma$, s.d.) $mol\,a^{-1}$ of $^{129}Xe(I)$. By comparison our past degassing rate integrated over the last 3.3 Ga would have been $8.1 \pm 3.9$ ($\pm 1\sigma$, s.d.) times higher than the present one. Taking a recent estimate of $450 \pm 50\,mol\,a^{-1}$ for the degassing of $^{3}He$ (ref. 41) leads to similar results with a past degassing rate $9.5 \pm 4.5$ ($\pm 1\sigma$, s.d.) higher than the present one. It is worth noting that the modern degassing rate was computed for the upper-mantle flux only, and whole mantle degassing might have played a major role in the past. A degassing rate up to 14 times the modern one is in agreement with elevated volumetric production rates inherent in convection models proposed for the early Earth (for example, ref. 42) and with degassing rates required to keep a significant portion of primordial $^{3}He$ in the Earth's mantle[43]. A higher degassing rate during the last 3.3 Ga is also consistent with a more convective mantle[44], sustained, for example, by a higher radioactive heat production from parent nuclides ($^{40}K$, $^{238-235}U$, $^{232}Th$) (for example, ref. 45).

Depletions in $^{134}Xe$ and $^{136}Xe$ for the primordial component similar to U-Xe as recorded by Barberton quartz may reflect either a mass-independent isotope fractionation process, not yet identified, or the presence of a nucleosynthetic anomaly (for example, r-process deficit) in the early atmosphere compared with other major components of the solar system (Chondritic or Solar Xe). This nucleosynthetic anomaly is problematic, as the contribution from meteorites, in particular the carbonaceous chondrites during the final stages of Earth's accretion is usually advocated to explain the abundances of volatile elements on Earth[10]. However, neither carbonaceous chondrites nor any other meteorite group contain U-Xe. Comets are primitive volatile-rich objects in the solar system that potentially may carry such an exotic primordial component, possibly inherited from other planetary systems formed in the vicinity of our Sun[46]. The abundances of the heavy noble gases in comets are not well constrained (only Ar has been measured so far[4]) and their capacity to transport noble gases will be highly dependent on the physical state of the ice (amorphous ice vs. clathrates). However, a rough estimate of the composition of these objects can be made (Supplementary Tables 4 and 5) based on experimental studies and on the results of the Rosetta space mission[4] (see Supplementary Discussion in Supplementary Information

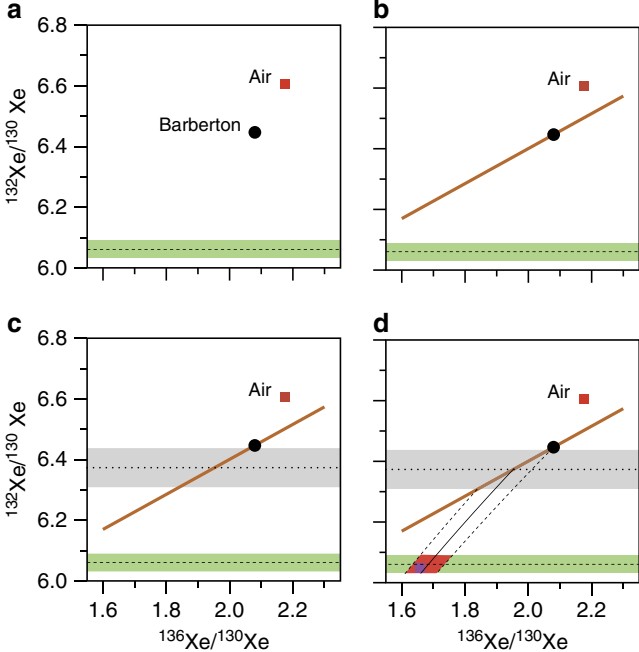

**Figure 7 | Schematic view of the successive steps leading to the determination of the initial $^{136}Xe/^{130}Xe$ for the ancient atmosphere trapped in Barberton quartz.** (**a**) The starting point uses only one data, the isotopic composition of Xe in Barberton quartz (the isotopic composition of the atmosphere is shown only for comparison) and makes the assumption that the starting $^{132}Xe/^{130}Xe$ is identical to SW-Xe (dashed line with the green range corresponding to the 2σ error of SW-Xe, ref. 33). (**b**) The orange line depicts the production of $^{132}Xe$ and $^{136}Xe$ by the spontaneous fission of $^{238}U$. Barberton quartz samples exhibit these excesses and must, therefore, lie on this line. (**c**) The dotted line and the grey range represents the initial $^{132}Xe/^{130}Xe$ for Xe in Barberton quartz before addition of fission products. It was obtained by applying the mass fractionation recorded on stable isotopes $^{126}Xe$ and $^{128}Xe$. (**d**) The intersection of the black line and the green range is used to estimate the primordial $^{136}Xe/^{130}Xe$ ratio for the Earth's atmosphere before the progressive mass-dependent isotopic fractionation occurred as recorded in Barberton quartz. The resulting space of possibilities (range at 2σ) appears in pink. The purple square corresponds to the isotopic composition of U-Xe (ref. 13).

and Supplementary Fig. 6). If Xe is present as amorphous ices in comets, then a 10% cometary contribution to a mass flux similar in magnitude to the Terrestrial Late Heavy Bombardment ($2 \times 10^{23}$ g, see Supplementary Information), added to the Earth following Moon formation, would be sufficient to bring the current budget of atmospheric Xe. For Xe contained in cometary clathrates, a 10% cometary contribution could deliver up to two orders of magnitude higher Xe than the surficial budget of Xe corrected for the loss (Supplementary Fig. 6). Thus, even if the nature of the late accreting events is not well constrained, comets may have contributed significantly to the budget of atmospheric noble gases, and specially Xe, after the Moon formed (Supplementary Fig. 6)[5].

The emerging picture for the history of Xe on Earth is schematically depicted in Fig. 10. The Earth accreted Chondritic heavy noble gases Xe (ref. 8) and Kr (ref. 7) still present in the modern mantle. Atmospheric Xe is not derived from Solar/Chondritic sources but from U-Xe, an exotic Xe component that may have been contributed by cometary bodies. The dichotomy for Xe between the primitive components stored in the mantle and in the atmosphere may thus be explained by distinct mixing for the two reservoirs: Solar/Chondritic for the Earth's mantle

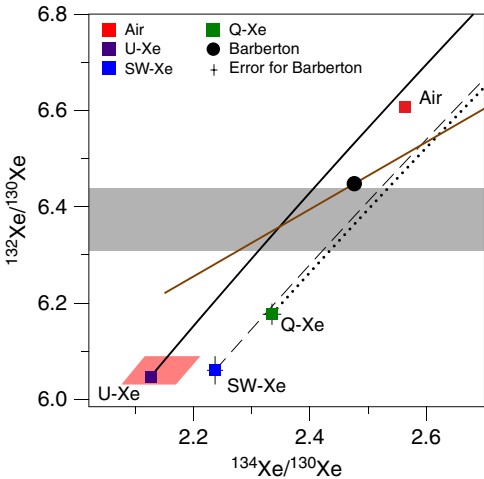

**Figure 8 | Three-isotope plot of Xe.** This figure demonstrates how the Archean atmospheric xenon trapped in Barberton quartz can only be produced by mass-related isotopic fractionation (black line) of a starting isotopic composition (pink area) similar to U-Xe (purple square) followed by the addition of xenon from the spontaneous fission of $^{238}U$ (brown line). Mass-dependent isotope fractionation (dashed and dotted lines) of SW-Xe (Solar Xe, blue square) and of Q-Xe (Chondritic Xe, green square) cannot lead to the isotopic compositions of Barberton or of the modern atmosphere. The grey area represents the range for the non-fissiogenic $^{132}Xe/^{130}Xe$ ratio of Barberton Xe obtained after propagation of the isotopic fractionation relative to SW-Xe measured on light Xe isotopes. Errors at 2σ.

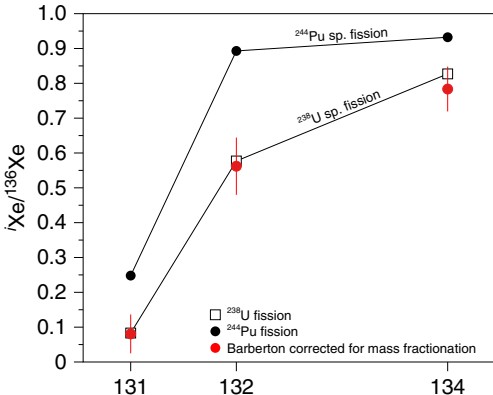

**Figure 9 | Fission spectrum of Barberton Xe corrected for mass-fractionation relative to a starting isotopic composition similar to U-Xe.** It corresponds to spontaneous fission of $^{238}U$. The fission spectra for fission of $^{238}U$ and $^{244}Pu$ are from a compilation in ref. 56. Errors at 2σ.

and Chondritic/Cometary for the atmosphere. The latter was already present 3.3 Ga ago. If this component was contributed by late bombardment events, these events must have preceded the formation of Barberton terranes, and could have occurred during the Late Heavy Bombardment of the Earth around 3.87 Ga ago, or earlier. At 3.3 Ga, atmospheric Xe had not reached its modern fractionated composition and subduction of ancient atmospheric Xe must have been limited to reconcile Xe data on mantle-derived samples[47] (Fig. 10a). The intense $^{129}Xe(I)$ degassing rate of $8 \pm 4$ ($\pm 1\sigma$, s.d.) mol a$^{-1}$ integrated over 3.3 Ga probably reflects degassing of the whole mantle in the active early Earth. The overall budget and isotopic composition of modern terrestrial xenon (Fig. 10b) have thus probably been shaped by various contributions of cosmochemical sources (Chondritic and possibly Cometary), atmospheric escape processes and complex

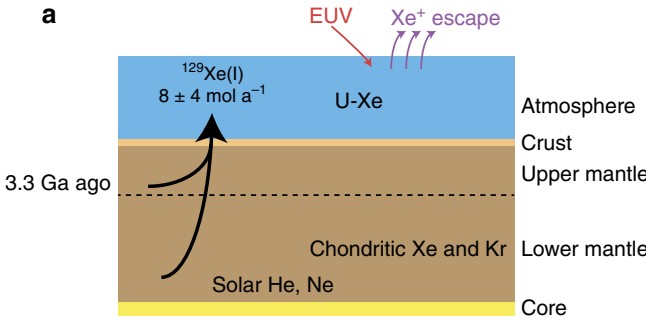

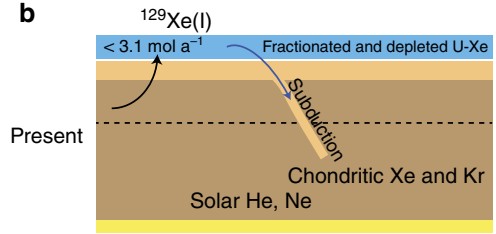

**Figure 10 | Schematic view of the history of terrestrial Xe.** Solar He and Ne (refs 6,57) and Chondritic Kr and Xe (refs 7,8) are present in the Earth's mantle since the time of Earth's accretion. The abundances and isotopic compositions of noble gases in the Earth's core remain unknown. (**a**) 3.3 Ga-ago, atmospheric Xe, derived from U-Xe, was still undergoing a progressive isotopic fractionation possibly from ionization by the EUV flux from the young sun (red arrow) and escape (purple arrows) processes[22,23]. Radiogenic $^{129}$Xe, produced by the decay of now extinct $^{129}$I ($T_{1/2} = 16$ Ma) was intensively ($> 8$ mol a$^{-1}$) degassed (thick black arrow) from the whole mantle. Subduction of atmospheric Xe must have been non-existent or limited (see text and ref. 47). (**b**) Terrestrial Xe on the modern Earth. Isotopic fractionation of atmospheric stopped. The degassing of radiogenic $^{129}$Xe is mainly via Mid-Ocean Ridge magmatism (thin black arrow) and atmospheric Xe is subducted back to the Earth's mantle. Subduction (deep blue arrow) probably reaches the lower mantle. Reservoirs are not to scale.

interactions between the different reservoirs on Earth. A precise determination of the time-dependent evolution of the isotopic composition of atmospheric Xe could shed light on major events on Earth such as the timing of the onset of Xe subduction and the mechanisms and cessation of atmospheric escape.

## Methods

**$^{40}$Ar-$^{39}$Ar experiment and determination of ages.** Separated quartz fractions of the samples have been analysed following the extended $^{40}$Ar-$^{39}$Ar method that enables simultaneous analysis of K, Cl abundances and the isotopic composition of argon[48].

Before irradiation, each sample was wrapped in aluminum foil. Samples were placed in a quartz tube, evacuated and sealed to a maximum length of 6.5 cm using a flame. Hb3gr hornblende samples, used as neutron flux monitors, were positioned at the bottom, and top of the quartz tube. Samples were irradiated (irradiation designated 'MN2014b') in the G-Ring In-Core Irradiation Tube facility of the TRIGA Reactor, Oregon State University. As the halogen-derived noble gas isotopes are produced by low energy thermal neutrons and epithermal neutrons, the irradiation cans were not Cd-shielded. Samples were irradiated for a few hour intervals each day over several weeks (22/4/14–1/7/14) to give a total irradiation time of 205 h for MN2014b.

The irradiation parameter $J$ is determined from the measured $^{40}$Ar/$^{39}$Ar ratio in the Hb3gr hornblende standards that were irradiated in the same tubes as the samples (equation 1):

$$J = \frac{(e^{\lambda t_m} - 1)}{^{40}Ar^* / ^{39}Ar} \tag{1}$$

where $t_m$ is the age of Hb3gr of $1074.9 \pm 3.5$ Ma (ref. 49) and $\lambda$ is the total decay constant ($5.531 \times 10^{-10}$ a$^{-1}$, ref. 50). Values of $J$ are $0.01789 \pm 0.00008$ and $0.01773 \pm 0.00008$ for Hb3gr monitors place at the top and bottom of the tube, respectively.

The abundances of Ca and Cl in samples can also be determined from Hb3gr using additional irradiation parameters, $\alpha$ (ref. 51) and $\beta$ (ref. 52) (equations 2 and 3):

$$\frac{K}{Ca} = \alpha \frac{^{39}Ar_K}{^{37}Ar_{Ca}} \tag{2}$$

$$\frac{K}{Cl} = \beta \frac{^{39}Ar_K}{^{38}Ar_{Cl}} \tag{3}$$

The abundances (in wt.%) of K, Ca and Cl in Hb3gr are, respectively, $1.247 \pm 0.008$, $7.45 \pm 0.09$ and $0.2379 \pm 0.0032$ (ref. 53). Analyses of Hb3gr yielded $\alpha$ values of $0.473 \pm 0.002$ (top) and $0.497 \pm 0.002$ (bottom). $\beta$ values are $1.271 \pm 0.018$ (top) and $1.263 \pm 0.019$ (bottom).

Thus in units of moles cm$^{-3}$ STP:

$$K = (3.66 \pm 0.03) \times \frac{[^{39}Ar]}{J} \tag{4}$$

$$Ca = (3.57 \pm 0.03) \times \frac{[^{37}Ar]}{J \times \alpha} \tag{5}$$

$$Cl = (4.04 \pm 0.04) \times \frac{[^{38}Ar]}{J \times \beta} \tag{6}$$

Following irradiation, samples were analysed in two successive steps: (1) step-crushing to release fluids trapped in fluid inclusions; (2) step-heating up to 1,700 °C to release K, Cl and Ar trapped in small inclusions, and present in the quartz lattice. Extraction, purification, measurement and correction techniques are already described elsewhere[54].

Results obtained for samples BMGA3-9, BMGA3-13 and BMGA3-3 are listed in Supplementary Table 1. The $^{40}$Ar/$^{36}$Ar ratios measured during the crushing and step-heating experiments are high (up to about 8,500) and cannot be solely explained by in situ decay of $^{40}$K even during 4.5 Ga. This $^{40}$Ar excess, referred to as $^{40}$Ar$_E$ hereafter, is correlated to the chlorine content (Figs 1 and 2) and probably linked to a hydrothermal fluid circulation through the samples[26]. One of the methods to correct the data for this $^{40}$Ar$_E$ contribution is to use the K-Cl-$^{40}$Ar (x-y-z) space where crushing and step-heating data would lie on a plane with the following equation (equation 7):

$$\frac{^{40}Ar}{^{36}Ar} = \left(\frac{^{40}Ar}{^{36}Ar}\right)_0 + A \times \left(\frac{Cl}{^{36}Ar}\right) + B \times \left(\frac{K}{^{36}Ar}\right) \tag{7}$$

where $\frac{^{40}Ar}{^{36}Ar}$, $\frac{Cl}{^{36}Ar}$ and $\frac{K}{^{36}Ar}$ are obtained during measurements (Supplementary Table 1), $\left(\frac{^{40}Ar}{^{36}Ar}\right)_0$ is the initial ratio trapped in the sample, $A$ represents the correlation between $^{40}$Ar$_E$ and the chlorine content, and $B$ ($= ^{40}$Ar*/K) reflects the relationship between in situ produced $^{40}$Ar ($^{40}$Ar*) and the potassium content computed from the abundance of $^{39}$Ar produced by neutron irradiation. The errors on the parameters of the plane defined by the data points were determined by a Monte Carlo propagation method using a Matlab code. First, for each point, coordinates on the x, y and z axes were divided by the mean error of the data set for normalization purpose. This was done to avoid overestimation of residuals resulting from different scale ranges (Supplementary Table 1). For each point a randomly generated cloud of 5,000 points was then created in order to properly represent the error envelop in the 3D space. The function of surface fitting 'sfit' of Matlab was applied to the 80,000 points. 'sfit' is a total least squares regression method through the entire cloud of points, the robust option permitting removal of outliers on an iterative basis based on least absolute residuals. Plots of residuals are shown in Supplementary Fig. 2. The fitting method led to a $^{40}$Ar/K value of $6.63 \times 10^{-5}$ ($\pm 8 \times 10^{-6}$, $\pm 2\sigma$, s.e.m.) for B that formally corresponds to an age of 3.3 ($\pm 0.1$) Ga ($2\sigma$, s.e.m.). ($^{40}$Ar/$^{36}$Ar)$_0$, representative of the initial $^{40}$Ar/$^{36}$Ar trapped in Barberton quartz, is $458 \pm 4$ ($2\sigma$, s.e.m.) higher than the present day atmospheric value of 298.6. This higher value is almost certainly due to some $^{40}$Ar excess remaining even after correction for $^{40}$Ar$_E$ linked to the chlorine content (Supplementary Fig. 3) and for radiogenic $^{40}$Ar from in situ decay of $^{40}$K during 3.3 Ga. The Matlab code used in this section is available through requests to G.A. (gavice@caltech.edu).

A second method originally proposed by Pujol et al.[20] has been applied to sample BMGA3-13 for which $^{40}$Ar$_E$ is less evidently linked to the Cl content but $^{40}$Ar/$^{36}$Ar measured during experiments are lower (see Fig. 1, Supplementary Fig. 3 and results in Supplementary Table 1). Crushing and step-heating results show a similar correlation in the $^{40}$Ar-Cl diagram (Fig. 1). A Cl\$^{40}$Ar$_E$ ratio of $6,500 \pm 949$ was thus obtained from analysis of crushing results. Second, this ratio was used to subtract $^{40}$Ar$_E$ using the chlorine content released during each heating step multiplied by Cl\$^{40}$Ar$_E$. A set of initial $^{40}$Ar/$^{36}$Ar ratios and times of fluid entrapments for in situ decay of $^{40}$K were then manually tested. A best solution was found for an initial atmospheric $^{40}$Ar/$^{36}$Ar ratio of $202 \pm 58$ ($2\sigma$, s.d.) for a fluid entrapment at $3.5 \pm 1.0$ Ga ($2\sigma$, s.d.) with a MSWD of 1.06. This age, although less precise, is in agreement with the age of $3.3 \pm 0.1$ Ga ($2\sigma$, s.e.m.) determined with the 3D correlation method applied to sample BMGA3-9. A more precise initial atmospheric ratio can be computed by taking an age of $3.3 \pm 0.1$ Ga ($2\sigma$, s.e.m.). It gives a value of $190 \pm 12$ ($2\sigma$, s.e.m.). It should be noted that error correlations are

not addressed in this section and thus, that the age uncertainty could be underestimated.

**Analytical procedure for Xe and Kr measurements.** Xenon and krypton isotopic compositions (and abundance of Xe) in fluid inclusions in quartz were determined by stepwise crushing (see results in Supplementary Tables 2 and 3). Before noble gas analyses, selected quartz fragments were gently crushed in a metal mortar to obtain grain-size fractions between 1 and 3 mm. This is the ideal range of sizes that minimizes adsorption of air on the surface of the grains and enables up to 2 g of sample to be loaded in to each crusher for noble gas analysis required to obtain a significant Xe signal in the mass spectrometer. Grains were subsequently cleaned with acetone in an ultrasonic bath, then rinsed with acetone and dried in an oven at 90 °C for 30 min. Following cleaning, quartz grains were hand-picked under a binocular microscope to ensure the absence of impurities on the surface of the grains and inside individual crystals. Each sample was then loaded in a stainless steel crusher. It consists of a modified valve where the valve's bellow has been replaced by a stainless tube moving downward when the modified valve is closed. Our samples were rich in fluid inclusions and their crushing released significant amounts of water and other inert or chemical reactive species (including $N_2$ and hydrocarbons). Ti-sponge getters, usually placed in a heating tube connected to the line and used to remove active species, were unable to remove all the water as indicated by monitoring the pressure in the purification system (up to $10^{-4}$ mbar). A new system was thus designed that consisted of an in-line Ti-sponge getter placed just after the crusher and heated at 700 °C ensuring that all the gas passed through the Ti-sponge. This new design solved the problem of the water purification, as demonstrated by the low pressure measured before the introduction in the mass spectrometer (for example, $1-5 \times 10^{-8}$ mbar). Xenon and krypton were condensed over a period of 20 min in a quartz tube held at liquid nitrogen temperature. Adsorption on the walls of the glass tube is unlikely to induce detectable isotopic fractionation[55]. The remaining part of the gas was pumped out. The amounts of $^{40}$Ar released during crushing of samples were so high, that even the minor fraction present in the tube with Xe and Kr prevented efficient ionization of Xe resulting in low sensitivity. Ten dilutions of the volume of the glass tube ($20\,cm^3$) into the whole line ($1{,}500\,cm^3$) dramatically decreased, by a factor of 750, the residual Ar partial pressure. Fractions rich in in Xe and Kr were then released and purified on three Ti-sponge getters at 550 °C for 5 min and at room temperature for 5 min before expansion into the noble gas multicollector mass spectrometer (Helix MC Plus, Thermo Fisher). Xe was the first gas to be analysed using a peak jumping mode with the magnet field, ion currents were detected using and a compact discrete dynode multiplier and Kr was subsequently analysed using a similar procedure. Procedural Xe blanks were monitored before each crushing experiment and were very low, on the order of $10^{-18}$ mol of $^{132}$Xe.

**Error propagation and compilation of the results.** Twenty-seven crushing experiments on seven distinct samples were conducted in total. Isotopic ratios of Xe released during each crushing experiment are shown in Supplementary Fig. 4. Very reproducible results were used to calculate an error-weighted average for each isotopic ratio (Supplementary Table 2) with MSWD values between 0.43 and 1.3. These values are satisfactory since MSWD values close to 1 indicate that errors are representative of the analytical uncertainty.

**Degassing rate inferred from low $^{129}$Xe(I) excess.** All errors given in this section are at $\pm 1\sigma$, s.d. The modern atmosphere contains $(4.06 \pm 0.05) \times 10^{12}$ mol of $^{129}$Xe (ref. 12) and an excess of $6.8 \pm 0.3\%$ of $^{129}$Xe(I) (ref. 35). This corresponds to $2.76 \pm 0.13 \times 10^{11}$ mol of $^{129}$Xe(I). Barberton $^{129}$Xe/$^{130}$Xe ratio has a $\delta^{129}$Xe$_{air}$ value of $6.3 \pm 1.3$‰ (Fig. 2a in the main text). Propagation of the isotopic fractionation computed on light isotopes (see above) toward $^{129}$Xe leads to a theoretical non-radiogenic delta value ($\delta^{129}$Xe$_{theor.}$) of $12.9 \pm 1.2$‰ for $^{129}$Xe. The difference of $6.6 \pm 1.8$‰ between theoretical and measured deviations relative to the isotopic composition of the atmosphere (Fig. 4) indicates that the 3.3 Ga-old atmosphere contained a $^{129}$Xe(I) excess of only $6.14 \pm 0.35\%$ corresponding to $(2.50 \pm 0.07) \times 10^{11}$ mol of $^{129}$Xe(I) with the conservative assumption that no Xe was lost from the atmosphere. This result enables calculation of a lower limit of $8 \pm 4$ mol a$^{-1}$ for the degassing rate of $^{129}$Xe(I) during the last 3.3 Ga.

**Data availability.** The authors declare that the data supporting the findings of this study are available within the paper and its Supplementary Information files.

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

## Acknowledgements

N. Arndt and A. Hofmann are gratefully acknowledged for providing samples. L. Zimmermann is thanked for technical mentorship and assistance. Y. Marrocchi, M. Kuga, L.C.P. Martin and R. Belissont are thanked for insightful discussions. This project was funded by the European Research Council under the European Community's Seventh Framework Program (FP7/2007-2013 grant agreement no. 267255 to B.M.) and by the French Ministère de l'Enseignement Supérieur et la Recherche (PhD funding to G.A.). R.B. acknowledges funding from UK NERC grant NE/M000427/1. This is CRPG contribution #2504.

## Author contributions

G.A. and B.M. collected the samples. G.A. and R.B. performed the experiments. G.A., R.B. and B.M. analysed the data and wrote the paper.

## Additional information

**Competing interests:** The authors declare no competing financial interests.

