## [Peer Review File · Nature Communications]

Reviewers' Comments:

Reviewer #1 (Remarks to the Author)

This is interesting work providing a valuable new constraint on the long-standing issues surrounding the Earth's atmospheric xenon abundance and isotopic signature. I believe it is ultimately worthy of publication in Nature Communications.

The authors report a new value for xenon isotopes from fluid inclusions trapped in the rocks from the Barberton Complex. They date the fluids using the Ar-Ar system. They identify mass fractionation in xenon isotopes relative to present-day air, and infer that the Earth's atmospheric xenon has become progressively more fractionated over time, confirming and adding confidence to previous results. They deduce that the proportion of radiogenic ^{129}Xe in the atmosphere was lower when these fluids were trapped, consistent with later addition of xenon by mantle degassing. They demonstrate that the underlying original composition was consistent with hypothetical U-Xe, identified as the starting composition for the modern atmosphere, rather than solar xenon. This suggests that the apparent U-Xe signature is not a mass-independent by-product of the fractionation process.

Before publication, however, some of the data treatment must be more rigorous and the discussion needs more clarity.

Some comments, starting with the data treatment...

1. The discussion of the plane fit needs to be improved. Three component mixing is hypothesised. If this is correct a plane will be a good fit to the data. If so, the hypothesis stands, and an Ar-Ar age and a trapped $^{40}\text{Ar}/^{36}\text{Ar}$ ratio can be derived as explained in the text. But...
 - a. I wasn't clear what was meant by "the results for Monte-Carlo simulations for..." in the caption to Fig 1 until I looked at Table S1. There are far more points on the Figure than in the Table. It's impossible to tell which Monte Carlo point relates to which release, and the reader doesn't get a very clear idea about the data from this figure. Please plot the data, not the simulated data, and give some idea of the residuals (e.g. plot the residual for each point in the supplement).
 - b. It isn't clear what the function "Fit" does (line 355). From outside research I find one that is an unweighted least squares fit, but is this it? Is there documentation somewhere.
 - c. I assume that most of the uncertainty in the measured ratios comes from the ^{36}Ar measurement (in Fig S3 the errors increase as ratios to ^{36}Ar increase). If so, this means that the errors are correlated in all the ratios. Was this taken into account when generating the point cloud (or in the fitting algorithm for the plane and for fig S3)?
 - d. This Monte Carlo approach doesn't overcome the problem that the fit is unweighted (if it is). If it is unweighted, in each Monte Carlo simulation some points have been given more weight than they should have and some have been given less. This will have an effect on the errors on the gradients and the intercept. The Monte Carlo has to be run on an appropriate fitting algorithm.
 - e. Finally, taking all that into account, was a plane a good fit to the data or not? (ie does the hypothesis of 3 component mixing stand?). This is particularly important given the elevated inferred $^{40}\text{Ar}/^{36}\text{Ar}$ ratio.
2. An alternative method was used for the other sample. In principle, given one dataset, all methods should yield identical (within rounding error) results. In practice, each method probably makes different implicit assumptions, which will lead to some difference in the eventual numbers. These should be explained. I can't think of a reason to adopt one for the first dataset and the other for the second; the data in S1 look broadly similar for the two datasets. So one approach should be adopted and the reason for the choice explained. The Pujol et al. approach at least reports an MSWD, though tracking the covariances through these calculations might be trickier than simply using an appropriate plane fit as for the first dataset.

3. Line 30. I suggest "simulations suggest that, during the final stages of solar system formation". It's a model that can account for some observations, not an established fact.
4. Line 56. Fig S1 cites 4 references, but the statement that refers to it in the text only cites 2. Is this right?
5. Line 60. Mass dependent fractionation is just a special case of mass independent fractionation. The statement seems to be that atmospheric loss might lead to fractionation that is mass independent, not a combination of dependent and independent (which doesn't seem to me to be a clear idea).
6. Line 77. This is what led me to examine the data reduction (comment 1). If there is an extra source of ^{40}Ar , why isn't there an extra source of xenon, and how can the agreement between the other sample and refs 23, 24 be considered significant (line 83)? It's dissatisfying to note an agreement with what is expected for one result while invoking some other process for one that disagrees.
7. Lines 86-103. Which isotope ratios were included in the fit, I assume ^{129}Xe was left out, what about ^{134}Xe and ^{136}Xe ? An MSWD of 4.1 (Fig. 2) says that the model doesn't account for the data. Quoting a gradient derived from this model with an error derived from the fit wouldn't be valid.
8. Lines 86-103. What other hypotheses were considered to account for the data? For instance, suppose the data have an excess in ^{128}Xe rather than a deficit in ^{129}Xe ? This would be hard to explain, but the apparent deficit in ^{124}Xe with the current model is even more difficult to explain and, by eye, looks to be just as significant as the deficit in ^{129}Xe that inferences are made from.
9. Lines 121-141. I think people will find this section confusing – I do. The chondritic and solar heavy isotope excesses relative to air have not been contributed by U and Pu fission (which first reading seems to suggest). It seems to say that these data are U-Xe, mass fractionated, with xenon from uranium decay added. But Pepin (doi:10.1023/A:1005236405730) says that the atmosphere contains U-Xe with Pu-derived xenon (not U-derived xenon) and herein it states that this sample is early atmosphere. So...is it proposed that Pu-derived xenon was degassed along with the addition of ^{129}Xe after this sample was isolated? If not, where is the Pu-derived xenon in this sample?
10. Lines 121-141 and suppl. Figs S6 and S7. S6 seems to show that the fission spectrum (presumably calculated over a starting composition) is U-derived. Fig S7 seems to show that, given it is U-derived, the starting composition is consistent with U-Xe. As written, there seems to be an element of circularity.
11. Please can you comment on how the Ar/Xe and Kr/Xe ratios compare to the modern atmosphere?

Reviewer #2 (Remarks to the Author)

The manuscript by Avicé and Marty presents fantastic new Xe isotope data in Archean quartz samples from the Barberton drill cores. The quartz samples trap atmospheric noble gases and therefore provide an opportunity to probe the composition of atmospheric noble gases in the Archean. The authors measure a Xe isotopic composition that shows Xe in modern air is mass fractionated by 12.9 per mil with respect to Xe in Archean air. The observation implies that the atmospheric Xe isotopic composition has been modified during the Hadean and Archean and that atmospheric loss was still ongoing at ~ 3.2 Ga. The result is important because it had long been assumed that Xe could only have been lost during Earth's accretion. The present data clearly demonstrate long term Xe loss from the atmosphere. Furthermore, they show that the Archean atmosphere cannot be derived from the solar or chondritic Xe but rather must be derived from a composition similar to U-Xe. Since the Xe in the mantle is either chondritic or solar, it implies that atmospheric Xe must be delivered at the very end of Earth's accretion. I think this is a nice and solid paper although not necessarily something that will fundamentally change our views on atmospheric evolution and on the formation of the atmosphere. That is because previous work by this group and has demonstrated that Archean Xe was less mass fractionated than modern day atmosphere and others (include Bob Pepin) have demonstrated that atmospheric Xe cannot be

derived from chondritic or solar Xe through mass dependent fractionation. Cometary noble gases have been invoked to deliver atmospheric noble gases most recently by Marty et al. (2016). Nonetheless, the data presented in this paper is of very high quality and will form a foundation that future studies will rely on. The manuscript is clearly written and arguments are well presented. For the most part, the appropriate literature has been properly cited. I do not have any substantial criticisms of the manuscript. I just have a few suggestions below that I hope will help the authors improve the paper.

Page 3: The uncertainties in Honda et al. (1986) are so large that it is impossible to distinguish between nebular Ne and solar wind implanted Ne (Ne-B), which is present in meteorites and solar nebular Ne. Therefore, for referencing the presence of solar end-member, I suggest using Yokochi and Marty (2004) who clearly demonstrated solar nebular Ne in the mantle.

2nd para on page 4: The authors should point out the reason for why the two different approaches were used for the two different samples. In the method sections of the manuscript I would suggest describing the Ar age dating technique in more detail. For example, after equation 9, the authors say that "the correlation between $^{40}\text{Ar}/^{36}\text{Ar}$ and the chlorine content...." I suggest adding a reference to fig S3 to show that there is a correlation with ^{40}Ar and Cl. However, I was not completely convinced that fig S3 demonstrates correlation of excess ^{40}Ar with Cl. If there was a correlation between Cl content and K content, wouldn't a correlation similar to that in Fig S3 be produced? The authors also indicate that they probably did not correct for excess Ar completely and that is why their initial $^{40}\text{Ar}/^{36}\text{Ar}$ is higher than modern day atmosphere. If that is correct, doesn't it also mean that they are underestimating the error in the sample age (since excess Ar is not properly accounted for)? These issues should be addressed.

On page 17, for the 2nd method, it is not immediately apparent that fluid inclusion and matrix Cl/ $^{40}\text{Ar}/^{36}\text{Ar}$ should be the same. Therefore, the authors should justify, why the crush data (fluid inclusions) can be used to correct for what is in the matrix. I assume they are using the relations seen in Fig S3. If yes, it should be clearly pointed out.

Degassing rate over 3.2 Ga; Based on the numbers, I calculate a mean value of 9.4 mol/yr of ^{129}Xe instead of 8 mol/yr. Please check.

For $^3\text{He}/^{130}\text{Xe}$ ratios, instead of Tieloff and Kunz, I suggest citing Moreira et al. (1998) who presents the popping rock data and calculates a value of 760 based on mantle $^{20}\text{Ne}/^{22}\text{Ne}$. Alternatively, use Mukhopadhyay (2012), who calculates a $^3\text{He}/^{130}\text{Xe}$ of 915 based on the modern day Xe isotopic composition of the mantle determined by Holland and Ballentine (2006). The authors should also comment a bit more on the fact that the average Xe degassing rates over the past 3.2 Ga are a factor of 18 higher than present day. In my opinion this is really high. How does this rate compare with other estimates based on noble gases (e.g., Coltice et al., 2009; Pujol et al., 2013; Gonnermann and Mukhopadhyay, 2009; Porcelli and Elliott, 2008)?

Page 3, 2nd to last line: add 'be' after 'may'.

Page 5 1st para line 2: replace with "permits a precise error.....to be computed"

Same para as above: Xenon in Barberton quartz.... Change 'has thus' to 'thus has'

Reviewer #3 (Remarks to the Author)

This study presents new high-precision Ar, Kr and Xe isotopic measurements of quartz-hosted fluid inclusions from the Barberton Greenstone Belt in South Africa. The data are of high quality and the reproducibility the authors have achieved in their Xe isotopic ratios is very impressive. The Xe results confirm that ancient Xe in the Archean atmosphere was characterized by a lesser degree of mass-dependent isotopic fractionation than is evident in the modern atmosphere, relative to

potential primordial compositions. The samples analyzed here are shown to have a younger age than samples previously measured (e.g., Pujol et al., 2009), and a further degree of mass fractionation relative to the primordial. The authors use this result to argue for progressive global mass fractionation of atmospheric Xe over time in the Archean. Based on a newly-resolved depletion in short-lived radiogenic ^{129}Xe , the authors derive a higher mantle outgassing rate in the past, which is a very exciting result. Furthermore, using a simple model, the authors are able to independently demonstrate that the "initial" Xe isotopic composition of the atmosphere was depleted in the heavy isotopes of Xe relative to any known primordial composition. The result presented here is consistent with the U-Xe initial composition previously determined based on the modern atmospheric composition (e.g., Pepin, 1991). Thus, the study presents a strong argument that the current "inventory" of planetary building blocks is incomplete – a major player for the origin of the atmosphere is still unknown. The authors conclude by speculating that comets may have delivered the U-Xe composition to the atmosphere.

Overall, the study builds a robust and compelling portrait of the time-evolution of Earth's atmospheric Xe composition, and makes valid points about volatile origins. It is a nice contribution and I recommend that it is published with minor revisions. Below are my specific comments.

SPECIFIC COMMENTS:

L 71-84: This paragraph describes Figure 1 and establishes the age of the fluid inclusions measured in this study. The 3D figure is not very easy to read – it might be better to additionally show either rotated views (viewing the plane of best fit edge-on), or subplots with the results collapsed in one dimension at a time ($^{40}\text{Ar}/^{36}\text{Ar}$ vs. $\text{K}/^{36}\text{Ar}$, $^{40}\text{Ar}/^{36}\text{Ar}$ vs. $\text{Cl}/^{36}\text{Ar}$, etc).

I am trying to understand the clustering in Figure 1. If I have interpreted the figure caption correctly, these are results for BMGA3-9, which has 7 crush steps and 11 heating steps. For each of the 18 steps, are 5000 points are randomly generated from within the error ellipsoid surrounding each of the 18 steps? Are those what the clusters are – essentially delineating the error space around the 18 steps? It would then make sense if the authors use the Matlab "fit" function to find the plane of best fit for all $(18 \times 5000) = 90,000$ simulated data points (if this is the case, fix line 355 where it says "fit" was applied to 5000 points).

There are a few different ways to estimate the uncertainty in the plane of best fit parameters. One way would be to have $\sim 100,000$ repeated fits of 18 values randomly drawn (with a normal distribution) from each error ellipsoid, and to gather statistics on the fit parameters from the 100,000 best fit results. However, given the nature of the dataset, this might give a huge overestimate of the uncertainty in the best fit plane and thus the age. So I think the method the authors use is fine with one amendment – since one of your dimensions is a few orders of magnitude smaller than the other two, it would be best to normalize the axes to make sure that residuals in K/Ar and Cl/Ar do not swamp out the residuals in $^{40}\text{Ar}/^{36}\text{Ar}$. Neglecting to do this could lead to the best fit algorithm tolerating very large residuals in the y-axis to preferentially fit the x and z dimensions. Normalizing each axis to a typical one sigma would mean that a 1s residual in x is given the same penalty as a 1s residual in y and z, so that would probably be the best normalization. If you do the fit this way, does it change the result significantly?

Why could the same method not be applied to BMGA3-13? The age determined using the method of Pujol et al. for BMGA3-13 is unfortunately very imprecise – I agree that it points to ancient gas, but the age from BMGA3-9 is much more compelling, so some comment on why this method could not be applied would be good. Also, since the authors argue that the initial $^{40}\text{Ar}/^{36}\text{Ar}$ from this sample is consistent with a lower $^{40}\text{Ar}/^{36}\text{Ar}$ than air determined based on previous measurements (Pujol et al., 2013), I think the supplementary figure (S3) should include an inset zoomed in on the intercept.

Were any of the other sample splits analyzed for Ar, Cl and K, or just these two? If it was just these two, somewhere in the manuscript it would be good to explain why these samples were

chosen for an age date while the others were not.

L 105 – 119: I think some re-ordering the paragraphs here might improve the manuscript. The radiogenic $^{129}\text{Xe}(\text{I})$ excess is computed relative to the mass fractionation fit for $^{126},^{128},^{130},^{131}\text{Xe}$, which is discussed in the next paragraph. I would move this outgassing paragraph later in the manuscript, or move some discussion of the mass dependent fractionation before it.

The present-day $^{129}\text{Xe}/^{132}\text{Xe}$ ratio of the mantle is just over 1 (would have been somewhat higher in the past). If ^{129}Xe outgassing shows up in the spectrum, shouldn't we expect outgassing of ^{132}Xe to matter as well? ^{132}Xe is trickier since there is in situ production after the gas was trapped. But if you use the non-fissiogenic ^{132}Xe from Figure 3 to derive a $\delta^{132}\text{Xe}_{\text{air,corr}}$, do you get a similar ballpark outgassing rate in the past?

L 121-141: Reading this paragraph is confusing – I would suggest some reshaping to make it very clear that the authors are not advocating for starting with solar wind, mass fractionating, then adding U-fission Xe to produce modern atmosphere. I know that's not what they're advocating! But a few times while reading it (L 124-126, L132) I did a double-take because I wondered whether they were. As I understand it, the authors are doing an inverse model based on constraints on the non-fissiogenic and primordial $^{132}\text{Xe}/^{130}\text{Xe}$, and the slopes of U-fission and mass fractionation in this space. So they are taking the measured Archean composition, attributing some portion of the heavy Xe to in situ U-fission since the inclusions were isolated, and then walking back the mass fractionation based on the light isotopes ($^{126},^{128},^{130},^{131}\text{Xe}$). Figures 3 and S7 show this nicely. The resulting initial composition space is depleted in $^{134},^{136}\text{Xe}$ relative to any primordial components (but ^{132}Xe is ok, as shown), and contains the U-Xe composition of Pepin.

It is nice that the authors can show this without using the modern atmospheric composition. It seems like a composition with a $^{136}\text{Xe}/^{130}\text{Xe}$ slightly higher than U-Xe is necessary to get perfect collinearity along the MDF line for modern atm, the Barberton-minus-U-fission and the initial – do the authors choose to show the black solid line shown in Figure 3 because they prefer the U-Xe value, or because they expect some contribution to the modern atmosphere from crustal outgassing, or something else?

L 131-132: The starting composition has a solar-like $^{132}\text{Xe}/^{130}\text{Xe}$, but it may be worth reiterating here that the starting composition is not solar – I believe the authors just need a target for the fission / mass fractionation inversion.

L 158-163: This is a nice conceptual model, and would benefit from a simple illustration – chondritic gas is in both the interior and surface reservoir, plus solar in the interior vs. plus cometary (or whatever is carrying the U-Xe signature) at the surface. That would provide a nice conceptual figure for people to cite.

Supplement L 129 – 166: The discussion of possible explanations for the depletion in ^{124}Xe relative to the mass-fractionation fit for $^{126},^{128},^{130},^{131}\text{Xe}$ is good and thorough. It's too bad that it cannot be explained as of yet -- the observation is puzzling.

Supplementary Figure S9: I am missing something here. When I read "corrected for missing Xe," I assume this means the corrected Xe/Kr is higher than the uncorrected Xe/Kr (inverting for the initial Xe/Kr before the mass-fractionating loss occurred). Why is the solid blue line higher than the dashed blue line?

SMALLER COMMENTS:

L 10: "heaviest noble gas" – this is probably an unnecessary addition, since readers may ask why radon is omitted

L 29: "too high to permit retention" – I would soften this. Temperatures were too high to permit significant retention, or something to that effect

L 31: I suggest making clear that the Grand Tack scenario is a hypothesis, since not everyone accepts that this is what happened.

L 39: radiogenic and fissiogenic

L 42-43: rephrase to make more clear – e.g., the Xe/Kr ratio in the Earth's atmosphere is depleted by a factor of 20 relative to chondrites

L 54-56: rephrase for clarity – perhaps isotopically fractionated to a lesser degree than modern atmosphere

L 93-95: for the benefit of readers unfamiliar with the sample type / potential signatures present in fluid inclusions, clarify that these argue against the presence of a mantle-derived component trapped within the inclusions.

L 107-110: reword this a little bit to make clear that although significant iodine is stored in the crust, ^{129}I was extinct by the time significant continental crust was accumulated, so mantle outgassing is the only source of radiogenic ^{129}Xe .

L 126-127: reword to make clear that ^{244}Pu was extinct at ~ 4.1 Ga – the mention of 3.2Ga could be confusing.

L 280-281: for total clarity, specify that this is in situ addition of U-fission Xe

Supplement L 136: change "decays in" to "decays to"

With best wishes,
Rita Parai

We provide a general response to the reviewers (R1 to R3) concerns regarding the methods
used to compute an age and initial argon isotope ratio for Barberton samples. The three
reviewers highlighted four main points, listed below, concerning the determination of the age
and the statistical treatment applied to the dataset:

- 1- details about the fit including scale normalization (R1 and R3)
- 2- correlations of errors (R1)
- 3- two different approaches on different samples (R1-3)
- 4- initial $^{40}\text{Ar}/^{36}\text{Ar}$ higher than in the atmosphere (R1-3)

In light of the reviewers' comments we have slightly revised the method used to determine the
age of the Barberton quartz samples and the initial $^{40}\text{Ar}/^{36}\text{Ar}$ ratio. The new method and the
results are described in more detail in our revised manuscript (L 92-117) and in the Methods
section (L 303-334). In detail, we respond to each point raised in the order listed above:

(1) As pointed out by reviewer 3, the z axis ($^{40}\text{Ar}/^{36}\text{Ar}$) scale range is much larger than those
of the x (Cl/ ^{36}Ar) and y (K/ ^{36}Ar) axes. Following this we have normalized each axis using
the mean error computed for the entire dataset. Using the normalized data, the obtained age is
3.3 ± 0.1 Ga (2σ) and the initial $^{40}\text{Ar}/^{36}\text{Ar}$ ratio is 458 ± 4 (2σ). These values were obtained by
applying the robust fit function of Matlab to fit a surface to a dataset ("sfit" :
<https://fr.mathworks.com/help/curvefit/sfit.html>).

(2) Reviewer #1 correctly points-out that because ^{36}Ar is the denominator in each term of the
23 ^{40}Ar -Cl-K 3D plot, the errors will be correlated. Ultimately all errors estimated from Ar-Ar
data are correlated as recently discussed by Vermeesch (2015), and it leads to complexity in
computing errors on the results since such a correlation requires to go back for example to
uncertainties on the irradiation parameters. Here we derived an age of 3.3 Ga and its
associated error without taking into account this error correlation. The age is also confirmed
by another approach that we wish to include in the revised version of our manuscript. It
consists in plotting the raw age given by the Ar-Ar method together with the K/Cl ratio of the
heating step or crushing step considered (Fig. 3). A low K/Cl ratio (high Cl) means a high
contribution of the hydrothermal component rich in ^{40}Ar excess. When the K/Cl ratio
decreases, ages also decrease to a range around 3.5-3.0 Ga; which is in agreement with the 3D
plotting method used to derive an age. Both approaches yield similar ages that are consistent
with previously reported isotopic ages for Barberton.

(3) The two different approaches used to interpret the Ar-Ar data are necessary because
samples differ in the extent to which excess ^{40}Ar is correlated with chlorine. This is reflected
in the much higher $^{40}\text{Ar}/^{36}\text{Ar}$ ratios for sample BMGA3-9 compared to sample BMGA3-13
(see Fig. below). Argon in BMGA3-13 is not well-correlated with Cl thus a 3D mixing
diagram does not include all Ar components for this sample and is therefore not appropriate to
determine an age. In contrast, the lower $^{40}\text{Ar}/^{36}\text{Ar}$ values of BMGA3-13 indicate that
comparatively lower excesses of ^{40}Ar are present, thus requiring lower correction.

(4) the high $^{40}\text{Ar}/^{36}\text{Ar}$ ratio remaining after correction for radiogenic and excess ^{40}Ar most
likely reflects the presence of an additional ^{40}Ar excess component less correlated with Cl.
Different origins for this component are possible. It might have been added to the fluid by
partial thermal diffusive loss of ^{40}Ar from rocks and minerals during fluid circulation, or it
could have resulted from leaching of ^{40}Ar from Cl-poor rocks. It is possible that the same
component is present in both quartz samples, it is just less dominant in BMGA3-13. We
emphasize that it was not the aim of our work to obtain a precise composition of the $^{40}\text{Ar}/^{36}\text{Ar}$
ratio (in contrast to the study by Pujol et al. 2013 in which an initial atmospheric $^{40}\text{Ar}/^{36}\text{Ar}$
ratio of 143 ± 24 was derived). The aim of our study was to obtain an accurate age to place the
Xe data in their correct context, for this reason we selected samples with the highest K
contents.

Our responses are in blue font.

Reviewer #1 (Remarks to the Author):

This is interesting work providing a valuable new constraint on the long-standing issues surrounding the Earth's atmospheric xenon abundance and isotopic signature. I believe it is ultimately worthy of publication in Nature Communications.

The authors report a new value for xenon isotopes from fluid inclusions trapped in the rocks from the Barberton Complex. They date the fluids using the Ar-Ar system. They identify mass fractionation in xenon isotopes relative to present-day air, and infer that the Earth's atmospheric xenon has become progressively more fractionated over time, confirming and adding confidence to previous results. They deduce that the proportion of radiogenic ^{129}Xe in the atmosphere was lower when these fluids were trapped, consistent with later addition of xenon by mantle degassing. They demonstrate that the underlying original composition was consistent with hypothetical U-Xe, identified as the starting composition for the modern atmosphere, rather than solar xenon. This suggests that the apparent U-Xe signature is not a mass-independent by-product of the fractionation process.

Before publication, however, some of the data treatment must be more rigorous and the discussion needs more clarity.

Some comments, starting with the data treatment...

1. The discussion of the plane fit needs to be improved. Three component mixing is hypothesised. If this is correct a plane will be a good fit to the data. If so, the hypothesis stands, and an Ar-Ar age and a trapped $40\text{Ar}/36\text{Ar}$ ratio can be derived as explained in the text. But...

a. I wasn't clear what was meant by "the results for Monte-Carlo simulations for..." in the caption to Fig 1 until I looked at Table S1. There are far more points on the Figure than in the Table. It's impossible to tell which Monte Carlo point relates to which release, and the reader doesn't get a very clear idea about the data from this figure. Please plot the data, not the simulated data, and give some idea of the residuals (e.g. plot the residual for each point in the supplement).

Fig. 1 has been modified and now shows only the data points. A plot of the residuals is added
as Supplementary Figure 2.

b. It isn't clear what the function "Fit" does (line 355). From outside research I find one that is
an unweighted least squares fit, but is this it? Is there documentation somewhere.

We applied the "sfit" function of Matlab. It is a total least square regression method
([https://fr.mathworks.com/help/curvefit/least-squares-](https://fr.mathworks.com/help/curvefit/least-squares-fitting.html?searchHighlight=robust%20fitting)
[fitting.html?searchHighlight=robust%20fitting](https://fr.mathworks.com/help/curvefit/least-squares-fitting.html?searchHighlight=robust%20fitting)) with a final function of the form $z = a*x +$
$b*y + c$.

c. I assume that most of the uncertainty in the measured ratios comes from the ^{36}Ar
measurement (in Fig S3 the errors increase as ratios to ^{36}Ar increase). If so, this means that
the errors are correlated in all the ratios. Was this taken into account when generating the
point cloud (or in the fitting algorithm for the plane and for fig S3)?

See our general response to the three reviewers.

106 d. This Monte Carlo approach doesn't overcome the problem that the fit is unweighted (if it
is). If it is unweighted, in each Monte Carlo simulation some points have been given more
weight than they should have and some have been given less. This will have an effect on the
errors on the gradients and the intercept. The Monte Carlo has to be run on an appropriate
fitting algorithm.

The "sfit" function provides an option to include weights on the fit. The weight is usually
taken as the inverse of the quadratic sum of errors on each coordinates. However, applying
this weight combined with the Monte Carlo approach leads to an unrealistic age around 4.1
114 Ga. This is probably because generating a cloud of points already leads to a large dispersion
of points for data with high uncertainties. Applying a least squares regression to such a cloud
corresponds to weighting the fit since the algorithm will easily fit points close to the surface
and ignore points (corresponding to large uncertainties) that are far away from the fitted
plane.

e. Finally, taking all that into account, was a plane a good fit to the data or not? (ie does the
hypothesis of 3 component mixing stand?). This is particularly important given the elevated
inferred $^{40}\text{Ar}/^{36}\text{Ar}$ ratio.

The r-squared value at the end of the fitting is 0.975. The elevated $^{40}\text{Ar}/^{36}\text{Ar}$ ratio probably

reflects that part of the ^{40}Ar excess has not been corrected and is thus translated into the
$^{40}\text{Ar}/^{36}\text{Ar}_0$ value. The model is consistent with fluid (trapped in fluid inclusions) that
interacted with the crust and became enriched in Cl, ^{40}Ar and, after trapping, in radiogenic
127 ^{40}Ar .

2. An alternative method was used for the other sample. In principle, given one dataset, all
methods should yield identical (within rounding error) results. In practice, each method
probably makes different implicit assumptions, which will lead to some difference in the
eventual numbers. These should be explained. I can't think of a reason to adopt one for the
first dataset and the other for the second; the data in S1 look broadly similar for the two
datasets. So one approach should be adopted and the reason for the choice explained. The
Pujol et al. approach at least reports an MSWD, though tracking the covariances through these
calculations might be trickier than simply using an appropriate plane fit as for the first dataset.
We explained the reasons of our choice in our general response to the three reviewers. To
briefly sum-up: i) the two datasets are different, BMGA3-9 has elevated $^{40}\text{Ar}/^{36}\text{Ar}$ ratios with
excess argon correlated to the chlorine content. The correlation is sufficiently well-defined so
that an age, in agreement with the geological context, can be computed. Excess argon, not
correlated with the chlorine content, prevents the determination of the initial isotopic
composition of atmospheric argon. BMGA3-13 has lower $^{40}\text{Ar}/^{36}\text{Ar}$ ratios but excess argon is
less correlated with the Cl content. The age is imprecise but by using an age of 3.3 ± 0.1 Ga
we can compute an initial $^{40}\text{Ar}/^{36}\text{Ar}$ ratio of about 190. This work was not focused on
searching the isotopic composition of paleo-atmospheric argon. In the main text we simply
note that this value is compatible with another study (Pujol et al., 2013).

3. Line 30. I suggest "simulations suggest that, during the final stages of solar system
formation". It's a model that can account for some observations, not an established fact.

Agree, it has been changed (L 35).

4. Line 56. Fig S1 cites 4 references, but the statement that refers to it in the text only cites 2.
Is this right?

Agree, it has been changed (L 418) main text is now citing the 4 references (Srinivasan, 1976,
Pujol et al. (2009, 2011, 2013)).

5. Line 60. Mass dependent fractionation is just a special case of mass independent

fractionation. The statement seems to be that atmospheric loss might lead to fractionation that
is mass independent, not a combination of dependent and independent (which doesn't seem to
me to be a clear idea).

Agree, it was not clear and has been changed to mass-dependent (L 75).

6. Line 77. This is what led me to examine the data reduction (comment 1). If there is an extra
source of ^{40}Ar , why isn't there an extra source of xenon, and how can the agreement between
the other sample and refs 23, 24 be considered significant (line 83)? It's dissatisfying to note
an agreement with what is expected for one result while invoking some other process for one
that disagrees.

Excess \$^{40}\text{Ar}\$ (probably linked to interactions of fluids with surrounding K-rich crustal rocks,
e.g. Kelley et al. (1986)) is here correlated to the Cl content and what we call "in-situ"
fissiogenic Xe from the decay of \$^{238}\text{U}\$ may have a similar origin. The elevated (\$^{40}\text{Ar}/^{36}\text{Ar}\$ )₀
derived for sample BMGA3-9 probably means that some excess \$^{40}\text{Ar}\$ is not correlated to Cl
and can thus hardly been corrected here. This excess \$^{40}\text{Ar}\$, if not due to decay of \$^{40}\text{K}\$ is likely
to end up in the initial (\$^{40}\text{Ar}/^{36}\text{Ar}\$ )₀. In that case it does not affect the age similarly to elevated
intercepts on the \$^{40}\text{Ar}/^{36}\text{Ar}\$ axis in \$^{40}\text{Ar}/^{36}\text{Ar}\$ vs. \$^{39}\text{Ar}/^{36}\text{Ar}\$ "classical" plots. In our first version
of the manuscript we labeled fissiogenic \$^{131-136}\text{Xe}\$ excesses "in-situ". This term is probably
inappropriate since potential readers could think that all Xe excesses have been produced in
the fluid inclusions during the 3.3 Ga of entrapment. We agree that part of these excesses,
could have been produced in the crust before entrapment. This does not change the fact that,
whatever the location of production (L 164-166), these excesses are due to the spontaneous
fission of \$^{238}\text{U}\$ so the conclusions of our study remain unchanged.

7. Lines 86-103. Which isotope ratios were included in the fit, I assume ^{129}Xe was left out,
what about ^{134}Xe and ^{136}Xe ? An MSWD of 4.1 (Fig. 2) says that the model doesn't account
for the data. Quoting a gradient derived from this model with an error derived from the fit
wouldn't be valid.

Unfortunately, there was a mistake in Fig. 2, the MSWD is 1.4 and not 4.1. It means that the
model can account for the data. \$^{126,128,130,131}\text{Xe}\$ isotopes have been included in the fit. It was
originally described in the supplementary information and has now been moved to the main
text (L 130-137).

8. Lines 86-103. What other hypotheses were considered to account for the data? For

instance, suppose the data have an excess in ^{128}Xe rather than a deficit in ^{129}Xe ? This would
be hard to explain, but the apparent deficit in ^{124}Xe with the current model is even more
difficult to explain and, by eye, looks to be just as significant as the deficit in ^{129}Xe that
inferences are made from.

196 ^{124}Xe is the second least abundant Xe isotopes. Its abundance is easily modified through
diverse nuclear reactions as explored in the Supplementary Information. ^{128}Xe is about 20
198 times more abundant than ^{124}Xe making this isotope less prone to these kinds of
199 production/destruction reactions. It leads us to consider that ^{128}Xe does not derive from so-
200 called nuclear excesses and that its overabundance compared to Air-Xe is due to the isotopic
fractionation of Archean atmospheric Xe. Furthermore, considering that ^{129}Xe is not depleted
and that ^{128}Xe is enriched (whatever the process) this leads to an estimated fractionation that
is poorly determined ($9.3 \pm 4.7 \text{‰} \cdot \text{u}^{-1}$ MSWD=4.3) together with an unexplained deficit in
204 ^{132}Xe .

9. Lines 121-141. I think people will find this section confusing – I do. The chondritic and
solar heavy isotope excesses relative to air have not been contributed by U and Pu fission
(which first reading seems to suggest). It seems to say that these data are U-Xe, mass
fractionated, with xenon from uranium decay added. But Pepin
(doi:10.1023/A:1005236405730) says that the atmosphere contains U-Xe with Pu-derived
xenon (not U-derived xenon) and herein it states that this sample is early atmosphere. So...is
it proposed that Pu-derived xenon was degassed along with the addition of ^{129}Xe after this
sample was isolated? If not, where is the Pu-derived xenon in this sample?

This section was indeed confusing as also pointed out by Reviewer 3. It has now been re-
written (L 181-185) to make it clear that, if Pu-Xe is present in the sampled Archean
atmosphere, it is probably masked by the important amount of ^{238}U -derived fissionogenic Xe
(either in-situ or from crustal fluids). For example, a simple calculation done with the
$^{134}\text{Xe}/^{136}\text{Xe}$ ratio gives a maximum of 2% of Pu-Xe ($0 \pm 2\%$ (2σ)) in the total fission
component (L182).

10. Lines 121-141 and suppl. Figs S6 and S7. S6 seems to show that the fission spectrum
(presumably calculated over a starting composition) is U-derived. Fig S7 seems to show that,
given it is U-derived, the starting composition is consistent with U-Xe. As written, there
seems to be an element of circularity.

Agree it has been re-written (L 177-179). Adding ^{238}U -derived fissionogenic Xe to a mass-

dependently fractionated U-Xe (the U-Xe from Pepin) appears to be the only reasonable way
to reproduce the isotopic composition of Xe measured in Barberton samples.

11. Please can you comment on how the Ar/Xe and Kr/Xe ratios compare to the modern
atmosphere?

Implicit in this question is the idea that the Xe/Kr ratios assist in, for example, validating and
further constraining the mechanism of Xe escape. Unfortunately we do not have sufficiently
precise measurements of the Ar/Kr/Xe ratios since the Xe, Kr and Ar-Ar experiments were
conducted separately. Furthermore, the variations of salinity, boiling and temperature could
have potentially modified the original trapped Ar/Kr/Xe ratio.

Reviewer #2 (Remarks to the Author):

The manuscript by Avice and Marty presents fantastic new Xe isotope data in Archean quartz
samples from the Barberton drill cores. The quartz samples trap atmospheric noble gases and
therefore provide an opportunity to probe the composition of atmospheric noble gases in the
Archean. The authors measure a Xe isotopic composition that shows Xe in modern air is mass
fractionated by 12.9 per mil with respect to Xe in Archean air. The observation implies that
the atmospheric Xe isotopic composition has been modified during the Hadean and Archean
and that atmospheric loss was still ongoing at ~ 3.2 Ga. The result is important because it had
long been assumed that Xe could only have been lost during Earth's accretion. The present
data clearly demonstrate long term Xe loss from the atmosphere. Furthermore, they show that
the Archean atmosphere cannot be derived from the solar or chondritic Xe but rather must be
derived from a composition similar to U-Xe. Since the Xe in the mantle is
either chondritic or solar, it implies that atmospheric Xe must be delivered at the very end of
Earth's accretion. I think this is a nice and solid paper although not necessarily something that
will fundamentally change our views on atmospheric evolution and on the formation of the
atmosphere. That is because previous work by this group and has demonstrated that Archean
Xe was less mass fractionated than modern day atmosphere and others (include Bob Pepin)
have demonstrated that atmospheric Xe cannot be derived from chondritic or solar Xe through
mass dependent fractionation. Cometary noble gases have been invoked to deliver
atmospheric noble gases most recently by Marty et al. (2016). Nonetheless, the data presented
in this paper is of very high quality and will form a foundation that future studies will rely on.

The manuscript is clearly written and arguments are well presented. For the most part, the
appropriate literature has been properly cited. I do not have any
substantial criticisms of the manuscript. I just have a few suggestions below that I hope will
help the authors improve the paper.

Page 3: The uncertainties in Honda et al. (1986) are so large that it is impossible to distinguish
between nebular Ne and solar wind implanted Ne (Ne-B), which is present in meteorites and
solar nebular Ne. Therefore, for referencing the presence of solar end-member, I suggest
using Yokochi and Marty (2004) who clearly demonstrated solar nebular Ne in the mantle.

Agree the reference has been changed.

2nd para on page 4: The authors should point out the reason for why the two different
approaches were used for the two different samples. In the method sections of the manuscript
I would suggest describing the Ar age dating technique in more detail. For example, after
equation 9, the authors say that “the correlation between $^{40}\text{Ar}/\text{E}$ and the chlorine content..... “
I suggest adding a reference to fig S3 to show that there is a correlation with ^{40}Ar and Cl.
However, I was not completely convinced that fig S3 demonstrates correlation of excess ^{40}Ar
with Cl. If there was a correlation between Cl content and K content, wouldn't a correlation
similar to that in Fig S3 be produced? The authors also indicate that they probably did not
correct for excess Ar completely and that is why their initial $^{40}\text{Ar}/^{36}\text{Ar}$ is higher than
modern day atmosphere. If that is correct, doesn't it also mean that they are underestimating
the error in the sample age (since excess Ar is not properly
accounted for)? These issues should be addressed.

The question about the two different approaches has been addressed in our general response
(see above). There is no evident correlation between the K and Cl contents (see Fig. below).
Excess Ar is not properly accounted for and ends up in the initial \$^{40}\text{Ar}/^{36}\text{Ar}\$. This excess argon
is thus not correlated to the K content and thus does not play a role in determining the age of
the sample.

On page 17, for the 2nd method, it is not immediately apparent that fluid inclusion and matrix
 Cl/40ArE should be the same. Therefore, the authors should justify, why the crush data (fluid
 inclusions) can be used to correct for what is in the matrix. I assume they are using the
 relations seen in Fig S3. If yes, it should be clearly pointed out.

Agree, it has been clarified that the relation seen in Fig. 2, 4 and Supplementary Fig. S3
 between argon excess and Cl in fluid inclusions is used to correct step-heating data for this
 component that has a characteristic $^{40}\text{ArE}/\text{Cl}$ ratio. Thus ^{40}Ar in situ + trapped = ^{40}Ar total -
 $\text{Cl} \cdot ^{40}\text{ArE}/\text{Cl}$. It has been clarified in the Methods section (L 324-327) as well as in the main
 text (L 107-111).

Degassing rate over 3.2 Ga; Based on the numbers, I calculate a mean value of 9.4 mol/yr of
 302 ^{129}Xe instead of 8 mol/yr. Please check.

The present abundance of $^{129}\text{Xe(I)}$ is $2.76 \pm 0.13 \times 10^{11}$ mol (corrected L 381), not 2.8 that
 gave indeed 9.4 mol/yr, 2.76×10^{11} mol leads to a degassing rate of 8 mol/yr.

For $^3\text{He}/^{130}\text{Xe}$ ratios, instead of Trieloff and Kunz, I suggest citing Moreira et al. (1998) who
 presents the popping rock data and calculates a value of 760 based on mantle $^{20}\text{Ne}/^{22}\text{Ne}$.
 Alternatively, use Mukhopadhyay (2012), who calculates a $^3\text{He}/^{130}\text{Xe}$ of 915 based on the
 modern day Xe isotopic composition of the mantle determined by Holland and Ballentine
 (2006). The authors should also comment a bit more on the fact that the average Xe degassing
 rates over the past 3.2 Ga are a factor of 18 higher than present day. In my opinion this is
 really high. How does this rate compare with other estimates based on noble gases (e.g.,
 Coltice et al., 2009; Pujol et al., 2013; Gonnermann and Mukhopadhyay, 2009; Porcelli and
 Elliott, 2008)?

We recalculated the Xe flux based on these new references (L201-203). It leads to a past
degassing rate between 5 and 14 times the modern one. This past degassing rate is considered
a lower limit since no Xe escape was taken into account. The error range for this estimate is
too large to place a firm constraint on the geodynamics for the last 3.3 Gyr however the high
integrated degassing rate is in agreement with most studies (van Thienen et al., 2005; Yokochi
and Marty, 2005).

Page 3, 2nd to last line: add 'be' after 'may'.

Page 5 1st para line 2: replace with "permits a precise error.....to be computed"

Same para as above: Xenon in Barberton quartz.... Change 'has thus' to 'thus has'

OK, changed.

Reviewer #3 (Remarks to the Author):

This study presents new high-precision Ar, Kr and Xe isotopic measurements of quartz-
hosted fluid inclusions from the Barberton Greenstone Belt in South Africa. The data are of
high quality and the reproducibility the authors have achieved in their Xe isotopic ratios is
very impressive. The Xe results confirm that ancient Xe in the Archean atmosphere was
characterized by a lesser degree of mass-dependent isotopic fractionation than is evident in
the modern atmosphere, relative to potential primordial compositions. The samples analyzed
here are shown to have a younger age than samples previously measured (e.g., Pujol et al.,
2009), and a further degree of mass fractionation relative to the primordial. The authors use
this result to argue for progressive global mass fractionation of atmospheric Xe over time in
the Archean. Based on a newly-resolved depletion in short-lived radiogenic ^{129}Xe , the
authors derive a higher mantle outgassing rate in the past, which is a very exciting
result. Furthermore, using a simple model, the authors are able to independently demonstrate
that the "initial" Xe isotopic composition of the atmosphere was depleted in the heavy
isotopes of Xe relative to any known primordial composition. The result presented here is
consistent with the U-Xe initial composition previously determined based on the modern
atmospheric composition (e.g., Pepin, 1991). Thus, the study presents a strong argument that
the current "inventory" of planetary building blocks is incomplete – a major player for the
origin of the atmosphere is still unknown. The authors conclude by speculating that comets
may have delivered the U-Xe composition to the atmosphere.

Overall, the study builds a robust and compelling portrait of the time-evolution of Earth's
atmospheric Xe composition, and makes valid points about volatile origins. It is a nice
contribution and I recommend that it is published with minor revisions. Below are my specific
comments.

SPECIFIC COMMENTS:

L 71-84: This paragraph describes Figure 1 and establishes the age of the fluid inclusions
measured in this study. The 3D figure is not very easy to read – it might be better to
additionally show either rotated views (viewing the plane of best fit edge-on), or subplots
with the results collapsed in one dimension at a time (40Ar/36Ar vs. K/36Ar, 40Ar/36Ar vs.
Cl/36Ar, etc).

We proposed a new figure (Fig. 4) following reviewer's (R1 and R3) comments. It contains
the plane obtained with the Monte-Carlo method, the initial dataset as well as edge-on view of
the plane in the Ar-Cl and Ar-K spaces.

I am trying to understand the clustering in Figure 1. If I have interpreted the figure caption
correctly, these are results for BMGA3-9, which has 7 crush steps and 11 heating steps. For
each of the 18 steps, are 5000 points are randomly generated from within the error ellipsoid
surrounding each of the 18 steps? Are those what the clusters are – essentially delineating the
error space around the 18 steps? It would then make sense if the authors use the Matlab “fit”
function to find the plane of best fit for all $(18 \times 5000) = 90,000$ simulated data points (if this
is the case, fix line 355 where it says “fit” was applied to 5000 points).

Agree, it has been fixed (L 312).

There are a few different ways to estimate the uncertainty in the plane of best fit parameters.
One way would be to have ~100,000 repeated fits of 18 values randomly drawn (with a
normal distribution) from each error ellipsoid, and to gather statistics on the fit parameters
from the 100,000 best fit results. However, given the nature of the dataset, this might give a
huge overestimate of the uncertainty in the best fit plane and thus the age. So I think the
method the authors use is fine with one amendment – since one of your dimensions is a few
orders of magnitude smaller than the other two, it would be best to normalize the axes to
make sure that residuals in K/Ar and Cl/Ar do not swamp out the residuals in 40Ar/36Ar.
Neglecting to do this could lead to the best fit algorithm tolerating very large residuals in the

y-axis to preferentially fit the x and z dimensions. Normalizing each axis to a typical one
sigma would mean that a 1s residual in x is given the same penalty as a
1s residual in y and z, so that would probably be the best normalization. If you do the fit this
way, does it change the result significantly?

This remark from reviewer three makes sense and we ran some tests to evaluate this method
(see our introduction in this response). Doing the normalization by a typical one sigma gives
results that are in agreement with our first method but the error is lower (around ± 0.1 Ga vs.
± 0.2 Ga). We thus adopted this normalization.

Why could the same method not be applied to BMGA3-13? The age determined using the
method of Pujol et al. for BMGA3-13 is unfortunately very imprecise – I agree that it points
to ancient gas, but the age from BMGA3-9 is much more compelling, so some comment on
why this method could not be applied would be good. Also, since the authors argue that the
initial $^{40}\text{Ar}/^{36}\text{Ar}$ from this sample is consistent with a lower $^{40}\text{Ar}/^{36}\text{Ar}$ than air determined
based on previous measurements (Pujol et al., 2013), I think the supplementary figure (S3)
should include an inset zoomed in on the intercept.

Our general response to reviewers included comments on the reasons why we used two
methods on two different samples. We included an inset zoomed on the intercept in
Supplementary Figure 3.

Were any of the other sample splits analyzed for Ar, Cl and K, or just these two? If it was just
these two, somewhere in the manuscript it would be good to explain why these samples were
chosen for an age date while the others were not.

Some other samples identical to those analyzed for Xe were also analyzed for Ar-Ar but,
unfortunately, either we have only results on crushing experiments or excess ^{40}Ar is poorly
correlated to Cl content (leaching of different crustal rocks?) so it was not possible to derive a
reasonable age (usually between 7 and 9 Ga (!) without ^{40}Ar correction). During the
preparation of this revised version, we made the choice here to include results from another
sample (BMGA3-3) because for elevated K/Cl ratios it gives a realistic age around 3.3 Ga.

L 105 – 119: I think some re-ordering the paragraphs here might improve the manuscript. The
radiogenic $^{129}\text{Xe}(I)$ excess is computed relative to the mass fractionation fit for
$^{126},^{128},^{130},^{131}\text{Xe}$, which is discussed in the next paragraph. I would move this outgassing

paragraph later in the manuscript, or move some discussion of the mass dependent
fractionation before it.

Agree, the paragraph about outgassing has been moved to the end of the discussion (LL 186-
213) just before the concluding remarks.

The present-day $^{129}\text{Xe}/^{132}\text{Xe}$ ratio of the mantle is just over 1 (would have been somewhat
higher in the past). If ^{129}Xe outgassing shows up in the spectrum, shouldn't we expect
outgassing of ^{132}Xe to matter as well? ^{132}Xe is trickier since there is in situ production after
the gas was trapped. But if you use the non-fissiogenic ^{132}Xe from Figure 3 to derive a
$\delta^{132}\text{Xe}_{\text{air,corr}}$, do you get a similar ballpark outgassing rate in the past?

It would have been very interesting to search for this ^{132}Xe outgassing. However, our
measurement of $\delta^{132}\text{Xe}$ encompasses the range determined for the isotopic fractionation
so that there is no ^{132}Xe excess left after correction for the isotopic fractionation.

L 121-141: Reading this paragraph is confusing – I would suggest some reshaping to make it
very clear that the authors are not advocating for starting with solar wind, mass fractionating,
then adding U-fission Xe to produce modern atmosphere. I know that's not what they're
advocating! But a few times while reading it (L 124-126, L132) I did a double-take because I
wondered whether they were. As I understand it, the authors are doing an inverse model based
on constraints on the non-fissiogenic and primordial $^{132}\text{Xe}/^{130}\text{Xe}$, and the slopes of U-
fission and mass fractionation in this space. So they are taking the measured Archean
composition, attributing some portion of the heavy Xe to in situ U-fission since the inclusions
were isolated, and then walking back the mass fractionation based on the light isotopes
($^{126},^{128},^{130},^{131}\text{Xe}$). Figures 3 and S7 show this nicely. The resulting initial composition
space is depleted in $^{134},^{136}\text{Xe}$ relative to any primordial components (but ^{132}Xe is
ok, as shown), and contains the U-Xe composition of Pepin.

A solar-like $^{132}\text{Xe}/^{130}\text{Xe}$ is only a starting hypothesis for deriving the initial $^{134}\text{Xe}/^{130}\text{Xe}$ and
$^{136}\text{Xe}/^{130}\text{Xe}$ ratios. We made some revisions (L155-160) to clarify that Solar-Xe (all isotopes)
is not the starting isotopic composition for the Earth's atmosphere.

It is nice that the authors can show this without using the modern atmospheric composition. It
seems like a composition with a $^{136}\text{Xe}/^{130}\text{Xe}$ slightly higher than U-Xe is necessary to get
perfect collinearity along the MDF line for modern atm, the Barberton-minus-U-fission and
the initial – do the authors choose to show the black solid line shown in Figure 3 because they

prefer the U-Xe value, or because they expect some contribution to the modern atmosphere
from crustal outgassing, or something else?

The black line was used here because U-Xe is the reference value. The error range for the
starting composition does not enable us to really conclude that it contains more ^{136}Xe than U-
Xe.

L 131-132: The starting composition has a solar-like $^{132}\text{Xe}/^{130}\text{Xe}$, but it may be worth
reiterating here that the starting composition is not solar – I believe the authors just need a
target for the fission / mass fractionation inversion.

OK, it has been clarified (L 160-162)

L 158-163: This is a nice conceptual model, and would benefit from a simple illustration –
chondritic gas is in both the interior and surface reservoir, plus solar in the interior vs. plus
cometary (or whatever is carrying the U-Xe signature) at the surface. That would provide a
nice conceptual figure for people to cite.

Agree, we now propose an illustration to our model in Figure 10.

Supplement L 129 – 166: The discussion of possible explanations for the depletion in ^{124}Xe
relative to the mass-fractionation fit for $^{126},^{128},^{130},^{131}\text{Xe}$ is good and thorough. It's too bad
that it cannot be explained as of yet -- the observation is puzzling.

Supplementary Figure S9: I am missing something here. When I read “corrected for missing
Xe,” I assume this means the corrected Xe/Kr is higher than the uncorrected Xe/Kr (inverting
for the initial Xe/Kr before the mass-fractionating loss occurred). Why is the solid blue line
higher than the dashed blue line?

An error was present both in the figure and in its caption. The solid blue line is indeed for
"corrected for missing Xe" and the dashed blue line for "uncorrected Xe". Figure and caption
have been corrected.

SMALLER COMMENTS:

L 10: “heaviest noble gas” – this is probably an unnecessary addition, since readers may ask
why radon is omitted

as well as Ununoctium, ok, changed.

L 29: “too high to permit retention” – I would soften this. Temperatures were too high to
permit significant retention, or something to that effect

OK, changed

L 31: I suggest making clear that the Grand Tack scenario is a hypothesis, since not everyone
accepts that this is what happened.

Agree, it has been clarified (L 35)

L 39: radiogenic and fissiogenic

OK

L 42-43: rephrase to make more clear – e.g., the Xe/Kr ratio in the Earth’s atmosphere is
depleted by a factor of 20 relative to chondrites

OK, rephrased (L 49)

L 54-56: rephrase for clarity – perhaps isotopically fractionated to a lesser degree than
modern atmosphere

OK, rephrased (L 67-68)

L 93-95: for the benefit of readers unfamiliar with the sample type / potential signatures
present in fluid inclusions, clarify that these argue against the presence of a mantle-derived
component trapped within the inclusions.

OK, changed and clarified (L 128-129).

L 107-110: reword this a little bit to make clear that although significant iodine is stored in
the crust, ^{129}I was extinct by the time significant continental crust was accumulated, so
mantle outgassing is the only source of radiogenic ^{129}Xe .

We agree that the major contribution of $^{129}\text{Xe}(\text{I})$ to the atmosphere is from mantle outgassing
and clarified this point (L 192-193). However, some works (e.g. Genda & Abe, 2005) pointed
out the potential presence of early oceans (before the giant impact) on Earth, potentially
containing some ^{129}I .

L 126-127: reword to make clear that ^{244}Pu was extinct at ~ 4.1 Ga – the mention of 3.2Ga
could be confusing.

Ok changed (L164-165).

L 280-281: for total clarity, specify that this is in situ addition of U-fission Xe

Clarified (L 165-167)

Supplement L 136: change “decays in” to “decays to”

OK, changed

With best wishes,

Rita Parai

Reviewers' Comments:

Reviewer #2 (Remarks to the Author)

The authors have addressed my comments and concerns successfully. Based on the revised manuscript, I have some minor comments that the authors should be able to address quite easily.

Figure 1: Please indicate explicitly how the data point in Fig 1 was computed. I assume that it is the error weighted average of all the measurements and is the composition listed in supplemental table 2? I would explicitly point that out in the main text or in the figure 1 caption, else it is not clear 'data from samples' is represented by a single point.

Line 58: The idea of CFF-Xe goes back to Meshik's work in the 1990s and is not recent. I suggest citing the original reference. The newer paper can be cited as an additional reference.

Statement starting on Line 61: I found this to be confusing and am not sure what the authors are trying to say. Please rephrase.

Line 232: Late Heavy bombardment may or may not be a real (see for example. Boehnke et al., 2016; PNAS). Even if it is real, the Xe data presented here does not put a time stamp on the timing of Xe delivery. I think the authors should say that it was delivered after the Moon formation associated with late accretion. Whether the delivered happened early during late accretion, continuously, or at the tail end of the late accretion is not constrained by the Xe data.

Line 255: Timing of onset of subduction and timing of onset of Xe subduction could be two different things. I think the authors should specify that its onset of Xe subduction they are referring to.

Line 311: What is do the authors mean when they say 'mean error of the dataset'? Is it standard error on the mean or the standard deviation? Please clarify. Usually, to scale everything to the same value, one subtracts the mean from individual data points and then divides by the standard deviation of the data set. Not clear if that is being done here.

Line 315: "It consists in a total least squares..." This appears to be either an odd phrasing or some grammatical error.

Line 327: Why is Ar_E less evidently linked to Cl for sample BMGA3-13? Visually, there still seems to be a fairly good correlation. To make this argument I think you need to show correlation coefficients. While there is scatter in BMGA3-13, there is also scatter in BMGA3-3 and to some extent in BMGA3-9.

Line 333 says initial Ar is 202 ± 58 for a fluid entrapment age of 3.5 ± 1.0 Ga, whereas main text says 190 ± 12 for ages between 3.2 and 3.4 Ga. I suggest making the initial Ar and the ages consistent in the main text and supplement.

Reviewer #3 (Remarks to the Author)

I have read through the revised manuscript, supplement and figures, and I feel that the revisions have suitably addressed the points raised in my review. This study builds a robust and compelling portrait of the time-evolution of Earth's atmospheric Xe composition, and makes valid points about volatile origins. It is a nice contribution and I recommend that it be published. Thanks.

Reviewer #4 (Remarks to the Author)

Review of the Ar/Ar age component of the manuscript "The origin and degassing history of the Earth's atmosphere revealed by Archean xenon"

The manuscript reports Ar/Ar analyses of vein quartz in an effort to determine the age of quartz veins from the Barberton sequence in South Africa, prior to reporting new Xe isotope measurements. All reviewers highlighted similar issues with the age component of the manuscript, that were summarised by the authors into 4 main points. The three reviews appear to be fair and thorough and raise similar issues. None are fatal criticisms but require serious effort. In this review I consider how the authors have addressed the 4 main points from the reviews.

The old ages of the heating steps are typical of this kind of material, demonstrating the presence of excess Ar. This is supported by the in vacuo crush data. Age correlation with Cl is not always present, though in this case it appears that similar Cl/ ^{40}Ar in the FI and quartz lattice in at least one samples is a reasonable conclusion from the data (Figs 2 and 3). Note that in the inset plots on Figure 4 the symbol colours (red and blue) of crushing and heating are reversed from those in Figs 2 and 3.

1. The new normalisation proposed by Rev3 is incorporated.

2. The correlation of errors in ($^{40}\text{Ar}/^{39}\text{Ar}$) dating is slowly developing as a concern, and will require major international effort to address it routinely in the future. Correlated uncertainties, in this case, mean that the age uncertainty given here is underestimated. The authors make no attempt to deal with the criticism. The demonstration (new Fig. 3) that the youngest heating step "age" is in the range 3-3.5 Ga, and they have high K/Cl, is not a particularly strong response as it does not deal with uncertainties. However, given that the calculated age is consistent with the long established age of the Barberton Group, and that the age precision is not absolutely crucial to the story, I suggest it remains as is but a statement to the effect that error correlation is not addressed is added to Methods.

3. That the 3-d age calculation method does not work for sample 3-13 (and several other samples, identified in response to review 3), clearly demonstrates the complexity of the distribution of Ar-Cl-K in these rocks. Although the second method - using the calculated Cl/ $^{40}\text{Ar}_E$ ratio to correct for non-atmospheric Ar - has been applied before (eg Pujol et al. 2015), in this case it generates such a large uncertainty as to make the calculated age pretty useless. All reviewers identified the need to undertake 2 age determination methods as an issue. I could see a case for removing the second method altogether, explaining why the first method only works in one case, followed by an explanation as to why the moderately precise age determined from sample 3-9 can reasonably be used as the age of all samples.

4. The complex distribution of Ar-Cl-K in these rocks is further evidenced by the high initial $^{40}\text{Ar}/^{36}\text{Ar}$ in sample 3-9 compared to the modern and, crucially, ancient atmosphere value. We are given only a vague statement of why this is. It would be useful for the authors to clearly state L322-324 what proportion of the total ^{40}Ar this represents, and how, if at all, it affects the age determination.

Please find below or point-by-point response to reviewers' comments. Our responses are in blue font.

Reviewer #2 (Remarks to the Author):

The authors have addressed my comments and concerns successfully. Based on the revised manuscript, I have some minor comments that the authors should be able to address quite easily.

Figure 1: Please indicate explicitly how the data point in Fig 1 was computed. I assume that it is the error weighted average of all the measurements and is the composition listed in supplemental table 2? I would explicitly point that out in the main text or in the figure 1 caption, else it is not clear 'data from samples' is represented by a single point. The blue data point for Barberton in Fig. 1 is, indeed, the result of the error-weighted correlation on 27 crushing experiments and obtained using the Isoplot 4.1 software (Ludwig, 1991). We clarified this point in the caption of Fig. 1 (L 568-569), in caption of Fig. 5 (L597) and it is also indicated in the main text (L139).

Line 58: The idea of CFF-Xe goes back to Meshik's work in the 1990s and is not recent. I suggest citing the original reference. The newer paper can be cited as an additional reference.

We agree with the reviewer. We now also cite (Meshik et al., 2000) at the point where CFF-Xe is discussed (L58-67) in our manuscript and we highlight additional references present in this 2000 paper (L60-61). Meshik et al. (2000) provide a comprehensive explanation of the nature of CFF-Xe as well as additional references to some original publications (from 90's and sometimes in Russian) from the same author or from the same group.

Statement starting on Line 61: I found this to be confusing and am not sure what the authors are trying to say. Please rephrase.

Here we are attempting to explain why invoking a SW-Xe component for the Earth's primary atmosphere is still problematic with or without CFF-Xe being involved. When SW-Xe is mass-fractionated in order to match the light Xe isotopes in the modern atmosphere (say ^{128}Xe for example), it leads to a $^{136}\text{Xe}/^{130}\text{Xe}$ ratio higher than in the modern atmosphere. While CCF-Xe does not have a fixed isotopic composition, it mainly contains fissionogenic $^{131-136}\text{Xe}$ isotopes and no stable ^{130}Xe for example. Thus, we do not exclude the intervention of CFF-Xe over Earth's history but any contribution this component (e.g. by mantle degassing) to the atmosphere, as suggested by Meshik et al (2016), will only compound the problem by further increasing the atmospheric $^{136}\text{Xe}/^{130}\text{Xe}$ value. Thus it is not possible to account for the composition of the early atmosphere by invoking only (mass fractionated) SW-Xe and CCF-Xe as primary

components We have expanded on this point and attempted to clarify it in the revised version of our manuscript (L61-67).

Line 232: Late Heavy bombardment may or may not be a real (see for example, Boehnke et al., 2016; PNAS). Even if it is real, the Xe data presented here does not put a time stamp on the timing of Xe delivery. I think the authors should say that it was delivered after the Moon formation associated with late accretion. Whether the delivered happened early during late accretion, continuously, or at the tail end of the late accretion is not constrained by the Xe data.

We agree that our work does not place constraints on the timing of cometary Xe to the Earth. Thus it remains unknown if this was coincident with the any putative Late Heavy Bombardment. Recently, Marty et al., (2016) identified the potential mass of volatile elements added to the Earth during the Late Heavy Bombardment and argued that it is possible to deliver the entire budget of surficial Ar without altering the D/H signature of the atmosphere. Furthermore, the Late Heavy Bombardment is thought to contain objects (comets?) originating from the outer regions of the solar system (see discussion in Marty et al. (2016)). We thus took this mass as a starting hypothesis for a Xe delivery by comets. As explained in the main text (L247-253) and in the Supplementary Information, the budget of heavy noble gases (Kr and Xe) in comets highly depends on the carrier phase (amorphous ice vs. clathrate).

We slightly re-phrase this part in this revised version of our manuscript to clarify that the Late Heavy Bombardment is only a possibility, and used here as an example for Xe delivery. We also emphasise that our results do not put constraints on its existence nor on the fact that this event is the one responsible for bringing volatiles to the Earth (L248-249, 253 and L166-168 in the Supplementary Information).

Line 255: Timing of onset of subduction and timing of onset of Xe subduction could be two different things. I think the authors should specify that its onset of Xe subduction they are referring to.

Agree, it has been changed (L273-274).

Line 311: What is do the authors mean when they say 'mean error of the dataset'? Is it standard error on the mean or the standard deviation? Please clarify. Usually, to scale everything to the same value, one subtracts the mean from individual data points and then divides by the standard deviation of the data set. Not clear if that is being done here.

To ensure that residuals on the three axis had comparable values we divided each coordinate for each point of our dataset by the average error of the dataset. We have checked, while preparing this revised version of our manuscript, by running all simulations again that taking the standard deviation of the data set gives identical results.

Line 315: "It consists in a total least squares..." This appears to be either an odd phrasing or some grammatical error.

Agree, we rephrased: " "sfit" is a total least squares regression method..."

Line 327: Why is Ar_E less evidently linked to CI for sample BMGA3-13? Visually, there still seems to be a fairly good correlation. To make this argument I think you need to show correlation coefficients. While there is scatter in BMGA3-13, there is also scatter in BMGA3-3 and to some extent in BMGA3-9.

In agreement with the reviewer's comment, we added R² values in Fig. 2 to demonstrate that Ar_E is less linked to CI for BMGA3-13 (R² = 0.95) than for samples BMGA3-3 and BMGA3-9 (R² = 0.98 together, or R²=0.98 and 0.99 for BMGA3-3 and BMGA3-9, respectively). This additional argument is also given in the main text (L126). In computing the correlation coefficient for BMGA3-3, we removed the first data point of measurements on BMGA3-3 (CI/³⁶Ar = 19, ⁴⁰Ar/³⁶Ar = 6000) since it appears to have a very low ⁴⁰Ar/³⁶Ar ratio for this given CI/³⁶Ar ratio compared to other measurements of the same sample.

Line 333 says initial Ar is 202±58 for a fluid entrapment age of 3.5 ± 1.0 Ga, whereas main text says 190 ± 12 for ages between 3.2 and 3.4 Ga. I suggest making the initial Ar and the ages consistent in the main text and supplement.

The initial ⁴⁰Ar/³⁶Ar ratio at 202±58 is indeed for the less constrained age of 3.5±1 Ga. Taking the more precise age determined with sample BMGA3-9 (3.3 ± 0.1 Ga) leads to initial values of 190 ± 12. We clarified this point (taking 190 ± 12 for an age of 3.3 ± 0.1 Ga) in the Methods section (L350-352).

Reviewer #3 (Remarks to the Author):

I have read through the revised manuscript, supplement and figures, and I feel that the revisions have suitably addressed the points raised in my review. This study builds a robust and compelling portrait of the time-evolution of Earth's atmospheric Xe composition, and makes valid points about volatile origins. It is a nice contribution and I recommend that it be published. Thanks.

Reviewer #4 (Remarks to the Author):

Review of the Ar/Ar age component of the manuscript “The origin and degassing history of the Earth’s atmosphere revealed by Archean xenon”

The manuscript reports Ar/Ar analyses of vein quartz in an effort to determine the age of quartz veins from the Barberton sequence in South Africa, prior to reporting new Xe isotope measurements. All reviewers highlighted similar issues with the age component of the manuscript, that were summarised by the authors into 4 main points. The three reviews appear to be fair and thorough and raise similar issues. None are fatal criticisms but require serious effort. In this review I consider how the authors have addressed the 4 main points from the reviews.

The old ages of the heating steps are typical of this kind of material, demonstrating the presence of excess Ar. This is supported by the in vacuo crush data. Age correlation with Cl is not always present, though in this case it appears that similar Cl/40Ar in the Fl and quartz lattice in at least one samples is a reasonable conclusion from the data (Figs 2 and 3).

Note that in the inset plots on Figure 4 the symbol colours (red and blue) of crushing and heating are reversed from those in Figs 2 and 3.

Thank you for noting this. We changed colours of the symbols in Figs 2 and 3 to maintain consistency.

1. The new normalisation proposed by Rev3 is incorporated.

Thanks, no response needed.

2. The correlation of errors in ($^{40}\text{Ar}/^{39}\text{Ar}$) dating is slowly developing as a concern, and will require major international effort to address it routinely in the future. Correlated uncertainties, in this case, mean that the age uncertainty given here is underestimated. The authors make no attempt to deal with the criticism. The demonstration (new Fig. 3) that the youngest heating step “age” is in the range 3-3.5 Ga, and they have high K/Cl, is not a particularly strong response as it does not deal with uncertainties. However, given that the calculated age is consistent with the long established age of the Barberton Group, and that the age precision is not absolutely crucial to the story, I suggest it remains as is but a statement to the effect that error correlation is not addressed is added to Methods.

We agree and added this statement on the absence of error correlation method in the Methods section (L352-353).

3. That the 3-d age calculation method does not work for sample 3-13 (and several other samples, identified in response to review 3), clearly demonstrates the complexity of the distribution of Ar-Cl-K in these rocks. Although the second method - using the calculated

Cl/ $^{40}\text{Ar}_E$ ratio to correct for non-atmospheric Ar - has been applied before (eg Pujol et al. 2015), in this case it generates such a large uncertainty as to make the calculated age pretty useless. All reviewers identified the need to undertake 2 age determination methods as an issue. I could see a case for removing the second method altogether, explaining why the first method only works in one case, followed by an explanation as to why the moderately precise age determined from sample 3-9 can reasonably be used as the age of all samples.

We acknowledge the reviewers' points and share some of the concern. It is important to establish the age of the fluids themselves because this provides direct evidence that it is Archean atmosphere contained in the fluid inclusions. Currently, ^{40}Ar - ^{39}Ar is the only isotopic dating technique capable of doing this. In such ancient samples, Ar is a mixture of one or more fluid types, ancient atmosphere and radiogenic ^{40}Ar . No two samples are likely to be comprised of exactly the same proportions of these component mixtures, so success is partly related to identifying (usually surreptitiously) samples with a higher proportion of the radiogenic ^{40}Ar component. Thus, it is not surprising that some samples will give more precisely defined ages than others, as the reviewer correctly notes in the comment. The other aspect is that there are different methods that can be applied to the ^{40}Ar - ^{39}Ar data to extract age information, our preferred method involves a 3-component mixing diagram to resolve the radiogenic component, we have used other approaches in the past, including correcting each data point individually for the presence of Cl-correlated excess ^{40}Ar . Ideally, both methods should result in the same age. Previous reviews requested the additional correction method based on Cl/ $^{40}\text{Ar}_E$, most likely because it helps build confidence that a self-consistent set of data have been obtained, which when treated in different ways yields reproducible results. Notwithstanding the large error on the age, we note that both methods also yield similar initial $^{40}\text{Ar}/^{36}\text{Ar}$ ratios. For the reasons outlined we prefer to keep both methods in the revised text.

4. The complex distribution of Ar-Cl-K in these rocks is further evidenced by the high initial $^{40}\text{Ar}/^{36}\text{Ar}$ in sample 3-9 compared to the modern and, crucially, ancient atmosphere value. We are given only a vague statement of why this is. It would be useful for the authors to clearly state L322-324 what proportion of the total ^{40}Ar this represents, and how, if at all, it affects the age determination.

The reviewer notes the issue of inter-sample variability, which relates to our response to the previous point. Initial $^{40}\text{Ar}/^{36}\text{Ar}$ ratios higher than atmospheric value are a common feature in ^{40}Ar - ^{39}Ar dating. This form of excess ^{40}Ar is incorporated into samples at the time of formation. In our component analysis we can discriminate the excess ^{40}Ar that is correlated to Cl, as elements both were added to the fluids during interaction with rocks in the crust, prior to being trapped as fluid inclusions in the quartz samples. Our 3-component analysis does not allow us to resolve other forms of excess ^{40}Ar unrelated to Cl, e.g. added by diffusive loss of radiogenic ^{40}Ar from crustal minerals.

These forms of excess ^{40}Ar components will have high $^{40}\text{Ar}/^{36}\text{Ar}$ above the Archean atmospheric value, but lack any relationship with K or Cl and on our Fig. 1 would plot along the $^{40}\text{Ar}/^{36}\text{Ar}$ axis. In our component plots any excess ^{40}Ar (not related to Cl) where present, and ancient atmospheric Ar form a single mixed component with intermediate $^{40}\text{Ar}/^{36}\text{Ar}$ values. We note this is a somewhat simplified technical explanation, however the main point is that regardless of the explanation for high initial $^{40}\text{Ar}/^{36}\text{Ar}$ value, it does not severely affect our ability to resolve the radiogenic ^{40}Ar component and hence the age determination.

References

- Ludwig, K.R., 1991. ISOPLOT; a plotting and regression program for radiogenic-isotope data; version 2.53. Open-File Report VL - USGS 91-445, 39.
- Marty, B., Avive, G., Sano, Y., Altwegg, K., Balsiger, H., Hässig, M., Morbidelli, A., Mousis, O., Rubin, M., 2016. Origins of volatile elements (H, C, N, noble gases) on Earth and Mars in light of recent results from the ROSETTA cometary mission. *EPSL* 441, 91–102.
- Meshik, A.P., Kehm, K., Hohenberg, C.M., 2000. Anomalous xenon in zone 13 Okelobondo. *Geochimica et Cosmochimica Acta* 64, 1651–1661.
doi:10.1016/S0016-7037(00)00334-3

Reviewers' Comments:

Reviewer #2:

Remarks to the Author:

I have read the authors response to my comments and re-read the manuscript. I am satisfied with the authors response and revisions. This is a nice contribution and needs to be published without further delay.

Reviewer #4:

Remarks to the Author:

The main issues highlighted in reviews have been adequately dealt with, I can recommend publication.